# EMT- and stroma-related gene expression and resistance to PD-1 blockade in urothelial cancer

Li Wang[1,2,3], Abdel Saci[4], Peter M. Szabo[4], Scott D. Chasalow[4], Mireia Castillo-Martin[5], Josep Domingo-Domenech[6], Arlene Siefker-Radtke[7], Padmanee Sharma[7], John P. Sfakianos[8], Yixuan Gong[9], Ana Dominguez-Andres[6], William K. Oh[9], David Mulholland[9], Alex Azrilevich[4], Liangyuan Hu[10], Carlos Cordon-Cardo [5], Hélène Salmon[11], Nina Bhardwaj[9], Jun Zhu [1,2,3,9] & Matthew D. Galsky[9]

Cancers infiltrated with T-cells are associated with a higher likelihood of response to PD-1/PD-L1 blockade. Counterintuitively, a correlation between epithelial–mesenchymal transition (EMT)-related gene expression and T-cell infiltration has been observed across tumor types. Here we demonstrate, using The Cancer Genome Atlas (TCGA) urothelial cancer dataset, that although a gene expression-based measure of infiltrating T-cell abundance and EMT-related gene expression are positively correlated, these signatures convey disparate prognostic information. We further demonstrate that non-hematopoietic stromal cells are a major source of EMT-related gene expression in bulk urothelial cancer transcriptomes. Finally, using a cohort of patients with metastatic urothelial cancer treated with a PD-1 inhibitor, nivolumab, we demonstrate that in patients with T-cell infiltrated tumors, higher EMT/stroma-related gene expression is associated with lower response rates and shorter progression-free and overall survival. Together, our findings suggest a stroma-mediated source of immune resistance in urothelial cancer and provide rationale for co-targeting PD-1 and stromal elements.

[1] Icahn Institute for Genomics and Multiscale Biology, Icahn School of Medicine at Mount Sinai, New York, NY 10029, USA. [2] Department of Genetics and Genomic Sciences, Icahn School of Medicine at Mount Sinai, New York, NY 10029, USA. [3] Sema4, A Mount Sinai venture, Stamford, CT 06902, USA. [4] Bristol-Myers Squibb, Princeton, NJ 08543, USA. [5] Department of Pathology, Icahn School of Medicine at Mount Sinai, New York, NY 10029, USA. [6] Departments of Oncology and Cancer Biology, Sidney Kimmel Cancer Center, Thomas Jefferson University, Philadelphia, PA 19107, USA. [7] Department of Genitourinary Medical Oncology, University of Texas MD Anderson Cancer Center, Houston, TX 77030, USA. [8] Department of Urology, Icahn School of Medicine at Mount Sinai, New York, NY 10029, USA. [9] Department of Medicine, Division of Hematology Oncology, Icahn School of Medicine at Mount Sinai, Tisch Cancer Institute, New York, NY 10029, USA. [10] Department of Population Health Science and Policy, Center for Biostatistics, Icahn School of Medicine at Mount Sinai, Tisch Cancer Institute, New York, NY 10029, USA. [11] Department of Oncological Sciences, Icahn School of Medicine at Mount Sinai, Tisch Cancer Institute, New York, NY 10029, USA. Correspondence and requests for materials should be addressed to J.Z. (email: jun.zhu@mssm.edu) or to M.D.G. (email: matthew.galsky@mssm.edu)

mmune checkpoint blockade has recently changed the treatment landscape for patients with metastatic urothelial cancer (UC). After several decades without significant therapeutic advances, clinical trials have demonstrated that durable responses are achieved in ~15–25% of patients with cisplatin-resistant metastatic UC treated with PD-1/PD-L1 blockade leading to regulatory approval of five distinct antibodies in the United States[1–6]. Because only a subset of patients benefit from treatment, there remains a critical need to understand mechanisms of intrinsic resistance.

Tumors infiltrated with T-cells, commonly referred to as "hot" tumors, are associated with a higher likelihood of response to immune checkpoint blockade[5–9]. These findings have led to the conceptual framework of "hot" vs. "cold" tumors as an approach to understanding mechanisms of sensitivity and resistance to treatment[10]. Although considerable emphasis has been placed on dissecting the immunobiology of "cold" tumors[11], a large proportion of patients with "hot" tumors also do not respond to PD-1/PD-L1 blockade, highlighting the need to better define resistance mechanisms in this latter group.

The biological process of epithelial–mesenchymal transition (EMT) involves epithelial cells assuming a mesenchymal phenotype, with enhanced capacity for invasion and metastasis. In studies encompassing a wide spectrum of malignancies, including UC, a positive correlation has been observed between T-cell infiltration and EMT-related gene expression[12–17]. The consistent association between EMT-related gene expression and T-cell infiltration has led to speculation regarding how EMT might impact the development of antitumor immunity and response to immune checkpoint blockade[18,19]. Indeed, some studies have suggested that patients with tumors with higher EMT-related gene expression should be more likely to benefit from immune checkpoint blockade[15,16] whereas others have linked EMT-related gene expression with immunotherapy resistance[20]. The seemingly counterintuitive relationship between EMT and T-cell infiltration, and contradictory clinical implications posed by prior studies, raise several critical questions: What is the cellular origin of EMT-related gene signatures derived from bulk UC transcriptomes? Does EMT-related gene expression indeed reflect the biological process of EMT? How do EMT-related gene expression and T-cell infiltration together impact outcomes in patients with UC treated with PD-1/PD-L1 blockade?

Here, using data from both TCGA and a cohort of UC patient-derived xenograft models, we provide support for a non-hematopoietic stromal source of EMT-related gene expression. Using data derived from a large clinical trial of patients with UC treated with the PD-1 inhibitor nivolumab we demonstrate that in patients with T-cell infiltrated tumors, higher EMT/stroma-related gene expression is associated with lower response rates and shorter progression-free and overall survival. Finally, we demonstrate that in T-cell infiltrated tumors with increased EMT/stroma-related gene expression, T-cells may be spatially separated from cancer cells. Together, our findings suggest a stroma-mediated source of immune resistance in UC and provide rationale for co-targeting PD-1 and stromal elements.

## Results

### EMT-related gene expression is associated with T-cell infiltration in UC in TCGA.
Gene expression of immune cell markers has been widely used to estimate blood cell components[21,22] and tumor infiltrating immune cell abundance[23,24]. We used a similar approach to estimate tumor-infiltrating T-cell abundance (ITA) in TCGA UC cohort (see Methods). Figure 1a shows 144 genes that were overexpressed in T-cells, and Fig. 1b shows the expression of the same 144 genes across UC tumor samples in TCGA.

We then searched for genes whose expression correlated with ITA and pathways enriched with these ITA correlated genes. The most highly enriched pathways positively correlating with ITA (Fig. 2a) included immune-related pathways, such as interferon, inflammatory and TNF pathways, as well as EMT (Molecular Signatures Database [MsigDB], hallmark EMT gene set[25]). EMT-related gene signature expression, calculated by average expression of 200 genes in the MsigDB EMT gene set, was also significantly correlated with ITA (Fig. 2b, Pearson's $\rho = 0.60$, $p$-value $< 1e-4$). Similar results were observed when using other EMT-related genes sets (see Supplementary Note 1 and Supplementary Fig. 1)[15,26].

Several groups have defined molecular subtypes of UC based on gene expression profiling (e.g., luminal and basal subtypes)[27–29]. We questioned whether EMT-related gene expression and ITA were enriched in specific molecular subtypes. We classified UC samples in TCGA ($n = 408$) into luminal, luminal-papillary, luminal-infiltrated, basal-squamous, and neuronal subtypes according to the most recent TCGA sub-classification[30]. Both EMT-related genes and ITA were most highly expressed in the luminal-infiltrated and basal-squamous subtypes (Fig. 2c, d), though EMT-related gene expression was slightly higher ($p = 0.047$ by two-sided Wilcoxon rank sum test) in the luminal-infiltrated (median = 10.23, $n = 78$) compared to the basal-squamous subtype (median = 10.00, $n = 142$) despite similar ITA (median = 5.54 and 5.55 for the two aforementioned subtypes, $p = 0.89$ by two-sided Wilcoxon rank sum test). The relationship between EMT-related gene expression and ITA within each subtype is shown in Supplementary Fig. 2.

### The positive correlation between EMT-related gene expression and ITA in UC is dependent on tumor purity.
Recent studies in colorectal cancer have revealed that genes comprising EMT signatures may be expressed predominantly from stromal cells rather than cancer cells[31–33]. Therefore, we next explored the relationship between EMT-related gene expression, ITA, and tumor purity as estimated by the computational tool ESTIMATE (Estimation of Stromal and Immune cells in Malignant Tumor tissues using Expression data)[23]. The ESTIMATE computational tool defines two gene signatures (referred to as immune_ESTIMATE and stromal_ESTIMATE, hereafter) to infer the proportion of the immune and stromal components from bulk transcriptomes and combines these individual components to estimate tumor purity. Using TCGA UC dataset ($n = 408$), we demonstrated that both ITA and EMT-related gene expression were highly positively correlated with lower tumor purity (Fig. 3a Spearman's $\rho = -0.85$ and Fig. 3b Spearman's $\rho = -0.84$, respectively). The strong positive correlation between EMT-related gene expression and ITA was no longer apparent (Fig. 3c, Spearman's $\rho = -0.23$) after accounting for tumor purity. EMT-related genes demonstrated a stronger positive correlation with stromal_ESTIMATE genes (Spearman's $\rho = 0.94$, Fig. 3d) than with immune_ESTIMATE genes (Spearman's $\rho = 0.68$, Fig. 3e) despite only a subset of genes in common between the gene sets (Fig. 3f). In contrast, ITA was more highly correlated with immune_ESTIMATE genes than with stromal_ESTIMATE genes (Supplementary Fig. 3). Together, these findings raised the possibility that EMT-related gene expression in UC may emanate from stromal cells in the tumor microenvironment rather than epithelial cancer cells.

### EMT-related gene expression and ITA have a disparate impact on survival in TCGA UC cohort.
We next explored the

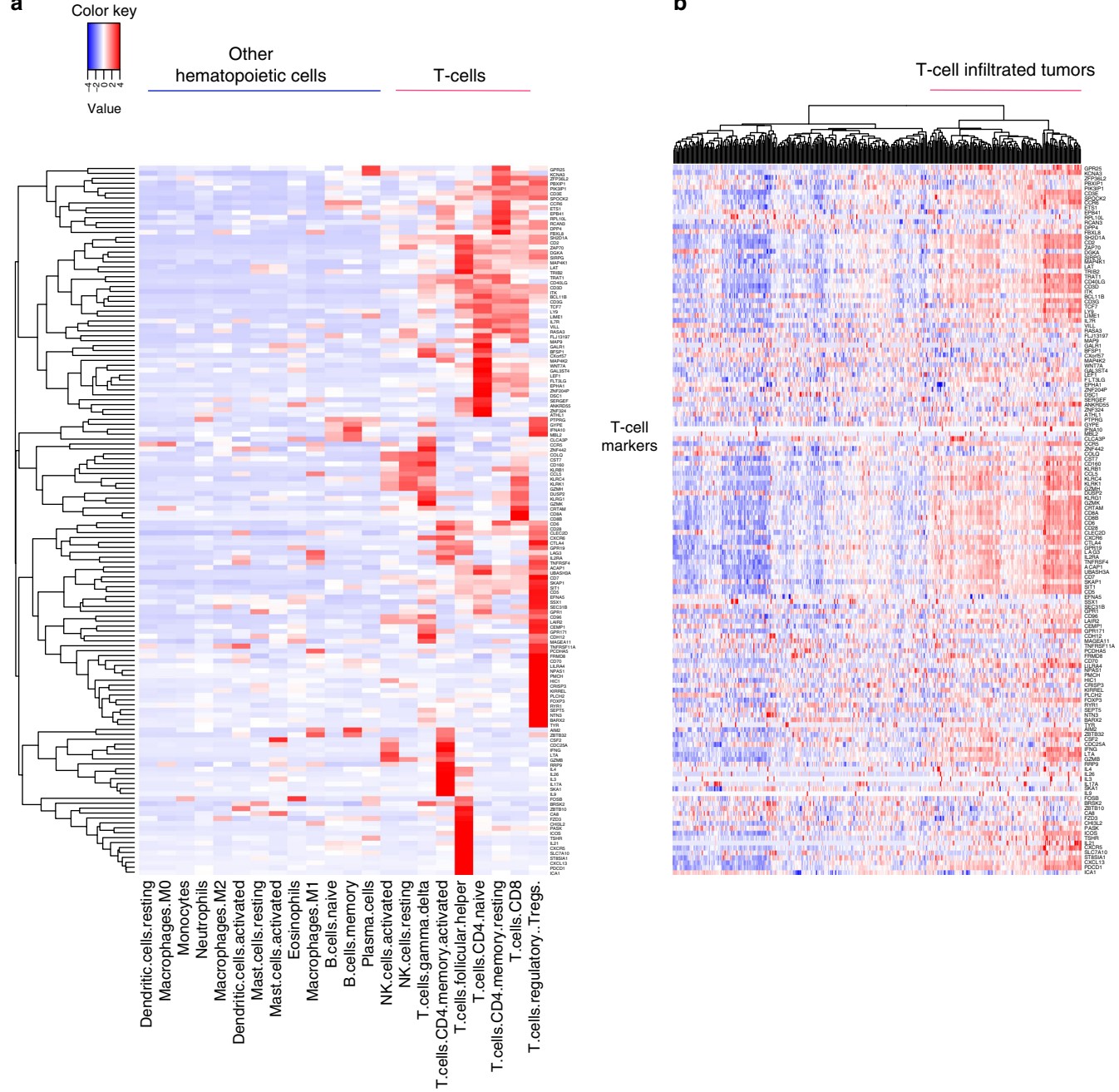

**Fig. 1** T-cell related gene expression is enriched in a subset of UC specimens. **a** Expression profiles of 144 T-cell marker genes across 22 different types or states of immune cells; **b** Expression profiles of the same 144 genes (the same order as in **a**) across 408 UC tumor samples in TCGA

prognostic significance of EMT-related gene expression and ITA in TCGA UC dataset, which is composed of patients with muscle-invasive UC of the bladder treated with radical cystectomy. Higher EMT-related gene expression was associated with worse overall survival (OS) in a univariable Cox regression model (Fig. 4a, HR = 1.45, Likelihood ratio test $X2 = 10.16$, $p = 0.0014$ when treated as a continuous variable), whereas ITA was not significantly associated with OS (Fig. 4b, HR = 0.84, Likelihood ratio test $X2 = 2.96$, p = 0.086 when treated as a continuous variable). As ITA and EMT-related gene expression were positively correlated, but demonstrated a potentially disparate impact on OS, we questioned whether combining these parameters would yield further prognostic and biologic insights. Indeed, the ratio of ITA to EMT gene expression was highly associated with OS in TCGA UC cohort (Fig. 4c, HR = 0.58, Likelihood ratio test $X2 = 28.16$, $p < 1e-4$ when treated as a continuous variable). Similarly, when both ITA and EMT-related gene expression were included additively in a bivariate Cox regression model, their impact on OS became more striking (HR = 2.08, Likelihood ratio test $X2 = 26.49$, $p < 1e-4$ for EMT and HR = 0.56, $X2 = 19.29$, $p < 1e-4$ for ITA in the bivariate Cox regression model when both were treated as continuous variables). The multiplicative interaction between ITA and EMT was not significant (Ratio of HRs = 0.85, Likelihood ratio test $X2 = 0.41$, $p = 0.52$) (Supplementary Fig. 4). As shown in Fig. 4d, when categorized into four groups based on median EMT and ITA gene expression, patients

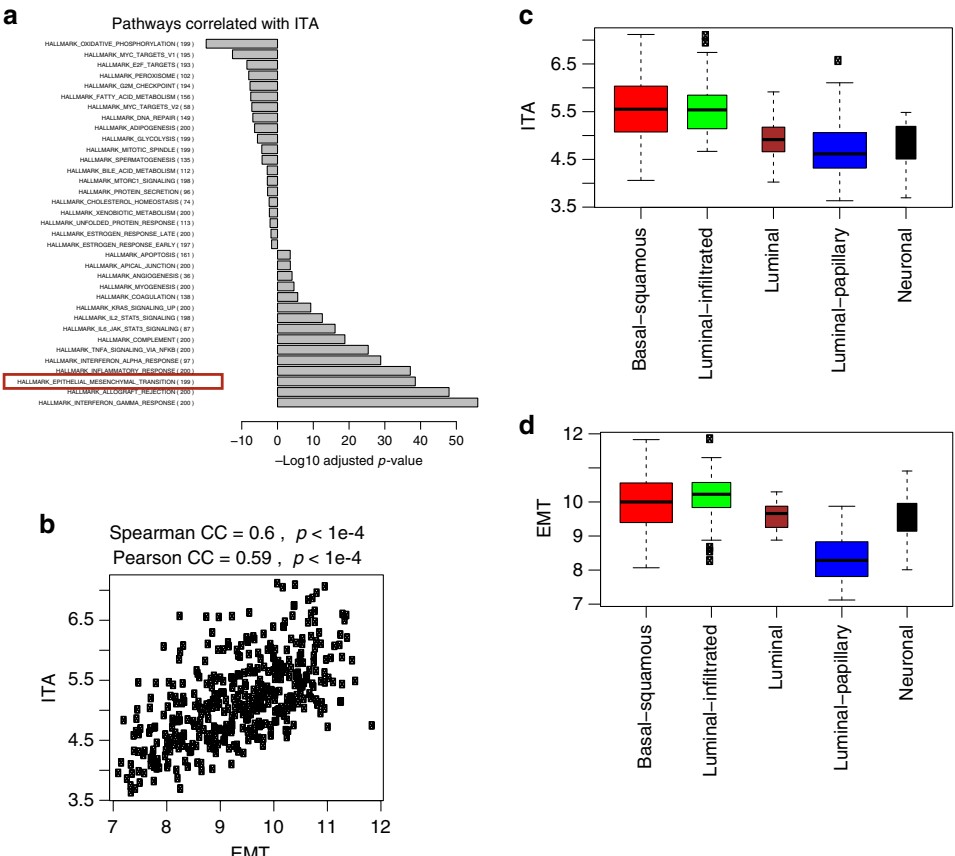

**Fig. 2** T-cell related gene expression and EMT-related gene expression are positively correlated in UC specimens. **a** Pathways ranked by their correlation with ITA. Pearson's correlation coefficient (CC) with ITA was calculated for each individual gene in TCGA UC datasets. Wilcoxon rank sum test was then used to compare CC values in each pathway with all the other genes. $X$-axis shows the −log10 $p$-value of the Wilcoxon test for each pathway (pathways that were negatively correlated with ITA were given log10 $p$-value instead); **b** Plot of the correlation between ITA and EMT in TCGA UC samples; **c** Plot EMT-related gene expression for different molecular subtypes of UC in TCGA; **d** Plot EMT-related gene expression for different molecular subtypes of UC in TCGA. For boxplots, boxes extend from the first to third quartiles, middle line shows median, whiskers extend to the most extreme data point which is no more than 1.5 times the interquartile range from the box, open circles show individual values that are more than 1.5 times the interquartile range from the box

with ITA$^{high}$ EMT$^{low}$ tumors demonstrated the best OS while patients with ITA$^{low}$ EMT$^{high}$ tumors demonstrated the worst OS.

Given that the impact of combined estimates of ITA and EMT-related gene expression on survival may potentially represent the balance of immune cells vs. immune-suppressive stromal elements in the tumor microenvironment, we considered whether gene expression-based estimates of specific T-cell subsets, or other immune cell types, might provide additional information (see Supplemental Methods). Conditioning on EMT-related gene expression in a bivariate Cox regression model, most immune cell types were associated with better OS, though T-cells and NK-cells were the most significantly associated with OS (Supplementary Table 1). Within T-cell subsets, gamma.delta, CD4 memory.resting, and CD8 T-cells were the most significantly associated with OS, though ITA performed similarly well.

Although EMT-related gene expression and stromal_ESTI-MATE genes were highly correlated, the EMT-related gene signature was more strongly associated with OS than the stromal_ESTIMATE gene signature. Conditioning on ITA and EMT, the stromal_ESTIMATE (average of stromal_ESTIMATE gene expression) signature was not significantly associated with OS (HR = 0.84, Likelihood ratio test $X2 = 0.17$, $p = 0.68$), whereas conditioning on ITA and stromal_ESTIMATE, the

EMT-related signature remained prognostic (HR = 2.38, Like-lihood ratio test $X2 = 5.54$, $p = 0.019$). When the prognostic significance of individual genes derived from both the EMT and stromal_ESTIMATE signatures were evaluated, 18 of the top 20 genes (Wald $p$-value < 1e−6, conditioning on ITA) most significantly associated with OS belonged to the EMT-related gene set and are subsequently referred to as the EMT/Stroma_core genes (Fig. 4e, Supplementary Table 2).

To determine if the prognostic impact of ITA and EMT-related gene expression was specific to UC or a more general phenomenon, we investigated the relationship between EMT-related gene expression, ITA, and OS in other types of solid tumors in TCGA. Similar to the case of UC, significant correlations were observed between ITA and EMT in the pan-cancer analysis, and the correlations were greatly reduced conditioning on the purity estimated by ESTIMATE (Supplementary Fig. 5). Further, the ratio of ITA to EMT was significantly associated with OS across a variety of tumor types (Supplementary Fig. 6, also see Supplementary Note 1).

**Stromal cells comprise a key source of EMT-related gene expression in UC patient-derived xenograft (PDX) models.** To further probe the source of EMT-related gene expression, we took advantage of a set of UC patient-derived xenograft (PDX)

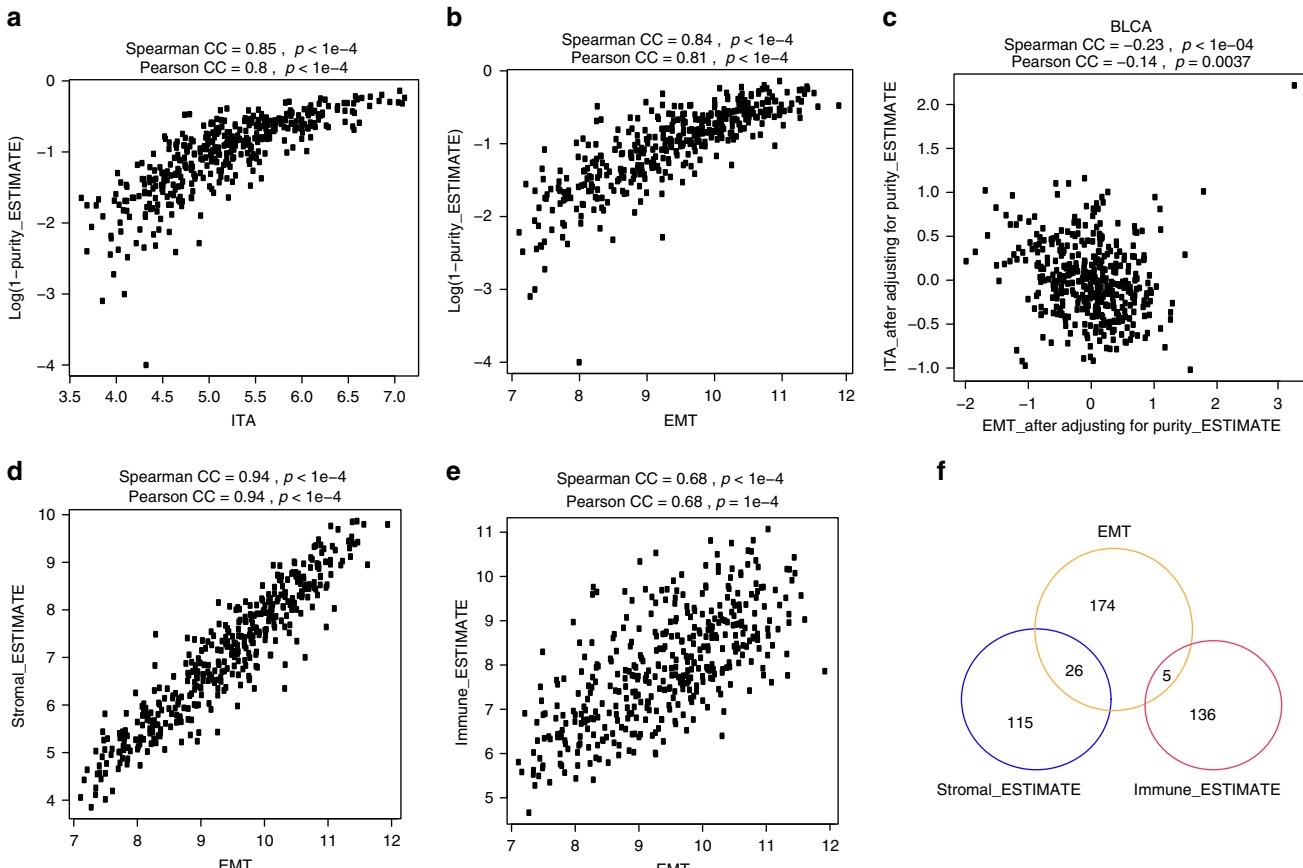

**Fig. 3** EMT-related gene expression is inversely correlated with tumor purity in UC specimens and likely emanates from stroma. Plot of correlation between **a** tumor purity and ITA, **b** tumor purity and EMT-related genes, **c** tumor purity adjusted ITA and EMT, **d** EMT-related genes and stromal signature (average of stromal_ESTIMATE gene expression), and **e** EMT-related genes and average of immune_ESTIMATE gene expression in TCGA UC samples; **f** Overlap among EMT-related, stromal_ESTIMATE and immune_ESTIMATE gene sets

models given that the transcriptome in these models is a mixture of human RNA (derived from cancer cells) and mouse RNA (derived from stromal cells). We analyzed RNA sequencing (RNAseq) data from five UC PDX models and used the Bamcmp algorithm[34] to separate RNAseq reads derived from mouse vs. human (see Methods). The median tumor purity as estimated by the fraction of total reads from human was 94% (Supplementary Fig. 7), which was higher than that of UC samples in TCGA (88% by ESTIMATE). At the individual gene level, EMT-related genes had a median of 30% reads mapped to mouse (Fig. 5a), which was significantly higher than that of all other genes (6%, two-sided Wilcoxon rank sum test statistic = 1,822,400, $p < 1e-4$) and lower than that of stromal_ESTIMATE genes (91%, two-sided Wilcoxon rank sum test statistic = 2756, $p < 1e0-4$, after the overlapping genes were removed). Dissecting the source of gene expression in bulk tumor specimens is not only a function of the relative proportion of reads from cancer vs. stromal cells but also the degree to which individual genes are expressed by these cellular compartments. Therefore, we also calculated log2 fold change of gene expression per mouse cell vs. human cell (as measured by species-specific reads-per-million (RPM), Fig. 5b). The median log2 FC for EMT-related genes was 2.02, which was significantly higher than that of all other genes, −0.47 (two-sided Wilcoxon rank sum test statistic = 1,958,000, $p < 1e-4$), and lower than that of stromal_ESTIMATE genes, 4.81 (two-sided Wilcoxon rank sum test statistic = 4247, $p < 1e-4$, after the overlapping genes were removed).

Finally, when the 18 EMT/Stroma_core genes were explored in our UC PDX models, 11 of 18 of these genes demonstrated a higher proportion of reads from mouse than human (Fig. 5c) and the median log2 fold change of gene expression per mouse cell vs. human cell was 4.23 (Fig. 5d). Together, these findings provide further evidence that stromal cells serve as a key source of EMT-related gene expression in UC.

**EMT-related (stromal) gene expression, T-cell infiltration, and their impact on response to immune checkpoint blockade and patient survival.** The anti-PD-1 antibody, nivolumab, has demonstrated durable responses in a subset of patients with metastatic UC in the phase II CheckMate 275 study, leading to regulatory approval in the United States and Europe for use in patients progressing despite platinum-based chemotherapy[2]. We used the CheckMate 275 dataset to query the impact of T-cell infiltration and EMT-related gene expression on objective response, progression-free survival (PFS), and OS in nivolumab-treated patients with metastatic UC. For this study, we did not use RNA-seq data to estimate the ITA and EMT signals. Rather, we used assays that potentially have a clearer path to clinical application. Targeted gene expression data from the EdgeSeq platform (HTG Molecular), and CD8 cell count based on immunohistochemistry (IHC), were generated from baseline archival tumor specimens. Of 270 patients enrolled, gene expression or CD8 IHC data were available from 217 and 263 patients, respectively; the final 'biomarker cohort' comprised 214 patients with both gene

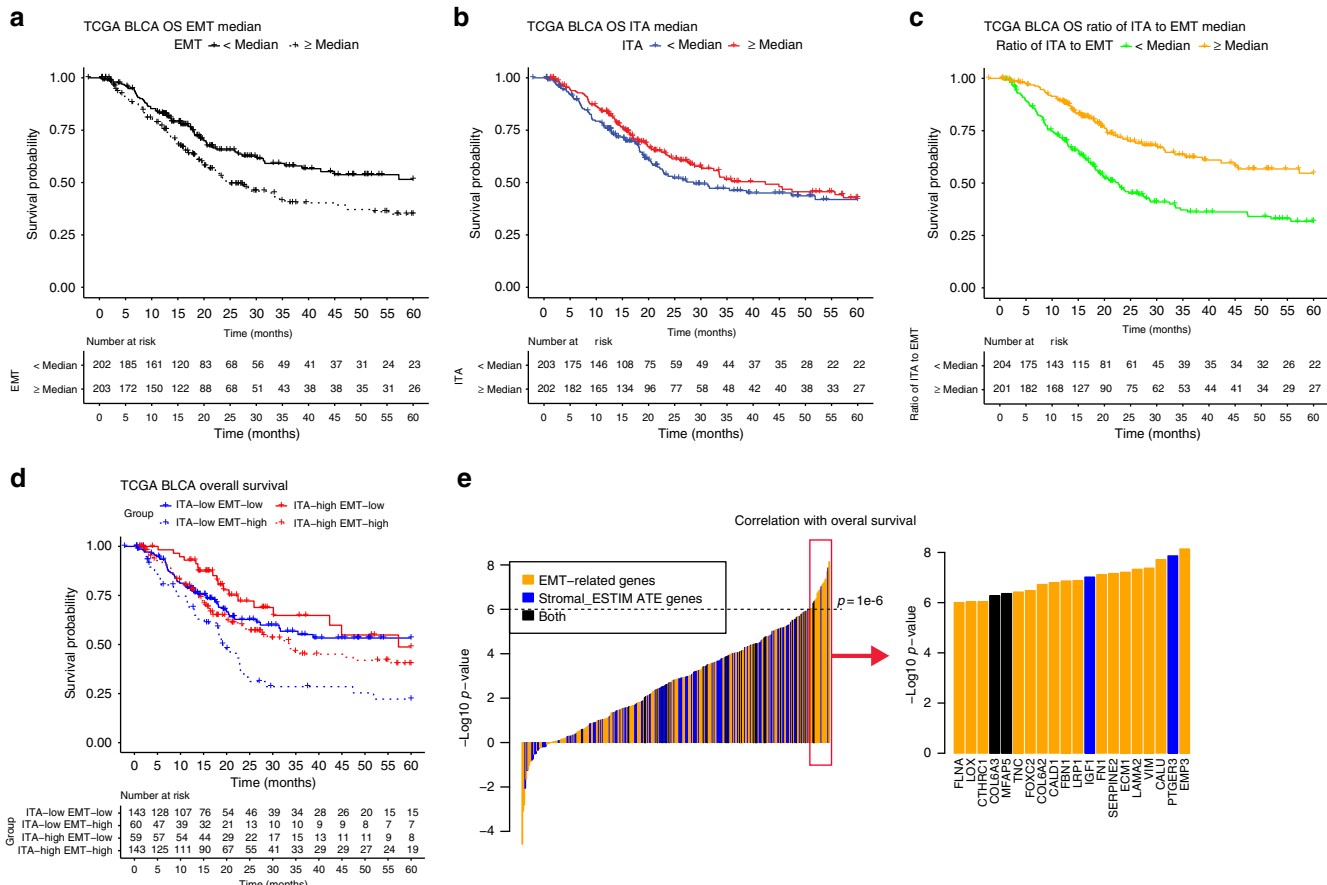

**Fig. 4** EMT-related gene expression and T-cell related gene expression confer disparate prognostic information in patients with muscle-invasive UC of the bladder treated with cystectomy. Kaplan–Meier survival curves for UC patients in TCGA ($n = 405$) divided into two groups by the **a** EMT-related gene expression, **b** ITA gene expression, **c** the ratio of ITA to EMT, or **d** divided into four groups by both EMT-related and ITA gene expression. Median EMT and ITA values were used to distinguish low vs. high expression. **e** Individual EMT-related and stromal_ESTIMATE genes were ranked according to the significance of their association with survival. Y-axis shows the −log10 p-value by the Wald test for each individual gene in the bivariate cox-regression model where ITA was included (those associated with worse survival were assigned log10 p-value instead)

expression and CD8 IHC data. The baseline characteristics of the biomarker cohort and overall CheckMate 275 cohort were very similar (Supplementary Table 3). Among the 200 genes in the EMT-related gene signature, 133 were included in the EdgeSeq expression panel. Among the 18 genes in the EMT/Stroma_core gene set, 8 were included in the EdgeSeq expression panel (*FLNA, EMP3, CALD1, FN1, FOXC2, LOX, FBN1,* and *TNC*). Results of analyses using the 8-gene EMT/Stroma_core signature were similar to results using the 133 EMT-related gene signature (data not shown). Therefore, we present results focused on the potentially more clinically tractable EMT/stroma_core signature.

In the CheckMate 275 cohort, CD8 expression by IHC and EMT/Stroma_core gene expression were positively correlated (Spearman's $\rho = 0.32$, $p < 1e-4$). From single-predictor models, greater CD8 infiltration was associated with a significantly higher objective response rate and longer PFS and OS, whereas EMT/Stroma_core gene expression alone was not associated with response rate, PFS, or OS (Table 1, Supplementary Table 4, Supplementary Fig. 8). However, when both CD8 IHC and EMT/Stroma_core gene expression were included in the model, a significant interaction was observed between the CD8 positive cell proportion and EMT/Stroma_core gene expression; that is, the negative association between EMT/Stroma_core gene expression and PFS, OS, or objective response depended on CD8 infiltration (Supplementary Fig. 4, 9, 10). This association was apparent at high, but not low, CD8 infiltration levels. The positive association

between CD8 infiltration and response, PFS, and OS increased in magnitude as EMT/Stroma_core gene expression decreased. Patients with high CD8 infiltration and low EMT/Stroma core gene expression had the highest response rates and longest PFS and OS, while patients with high CD8 infiltration but high EMT/ Stroma core gene expression had worse outcomes. The CD8: EMT/Stroma_core interaction term remained significant for PFS, OS, and objective response even when other baseline variables including hemoglobin, PD-L1 expression (as measured by IHC on cancer cells), and the presence of liver metastases were included in the model (Table 1, Supplementary Table 4). For illustrative purposes, CD8 infiltration and EMT/Stroma_core gene expression were dichotomized at median expression levels to generate four patient subgroups with objective response rate, PFS, and OS by subgroup shown in Fig. 6a–c. Together, these findings suggest that in CD8-infiltrated UC, EMT/Stroma-related gene expression is associated with resistance to PD-1 blockade.

We hypothesized that CD8-infiltrated tumors with high EMT/ Stroma core gene expression might represent the previously described "immune excluded" phenotype with CD8 cells spatially separated from cancer cells and restricted to stromal regions[35]. Therefore, we performed an exploratory analysis of the spatial localization of CD8 cells in a subset of CD8-infiltrated specimens with higher vs. lower EMT/Stroma_core gene expression from the CheckMate 275 cohort. A genitourinary pathologist (M. C-M.) blinded to the EMT/Stroma_core gene expression scores

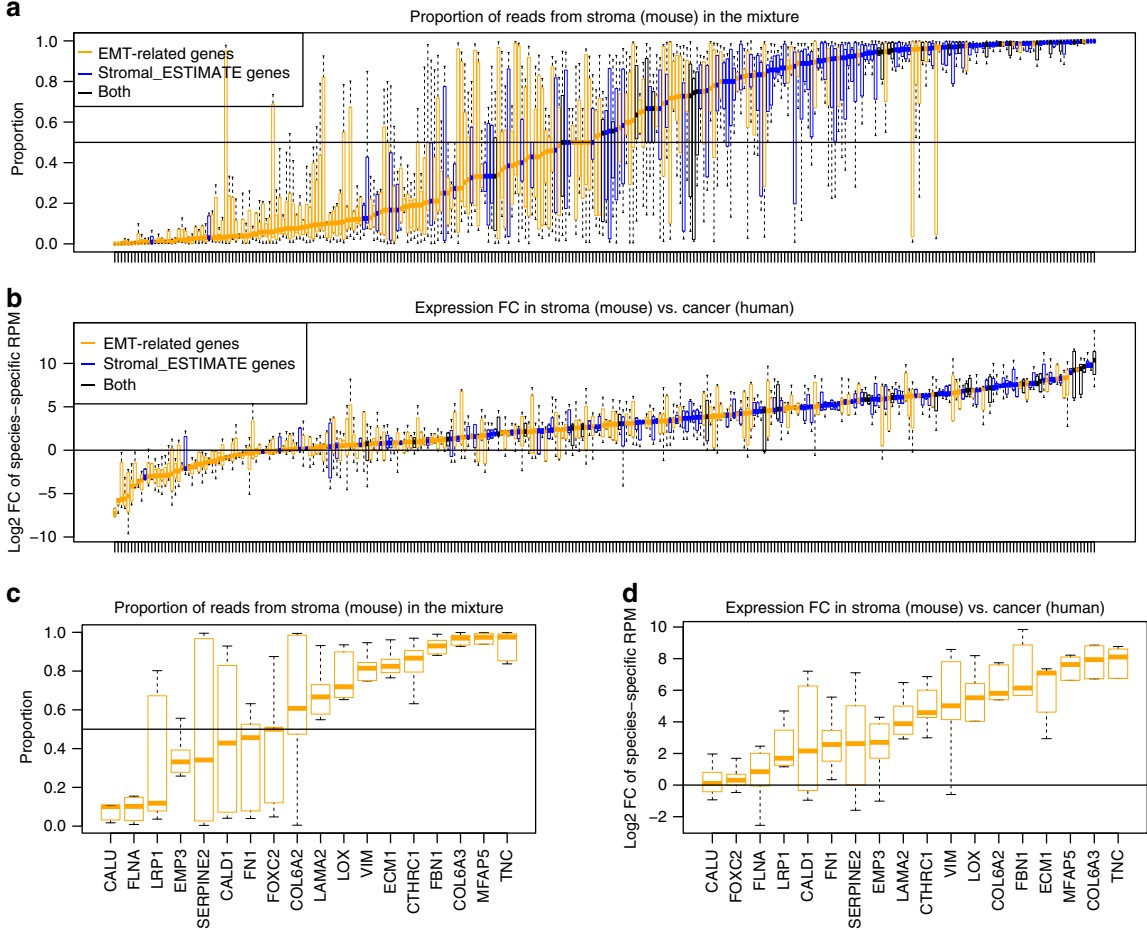

**Fig. 5** EMT-related gene expression emanates predominantly from a stromal source in UC PDX models. **a** Fraction of reads mapped to mouse for each EMT-related and stromal signal gene; **b** Fold change of the species-specific RPM value between human and mouse for each EMT-related and stromal signal gene; **c** Fraction of reads mapped to mouse for the top 18 EMT genes correlated with survival (EMT/Stroma_core genes); **d** Fold change of the species-specific RPM value between human and mouse for the top 18 EMT-related genes (EMT/Stroma_core genes) correlated with survival. For boxplots, boxes extend from the first to third quartiles, middle line shows median, whiskers extend to the most extreme data point which is no more than 1.5 times the interquartile range from the box, open circles show individual values that are more than 1.5 times the interquartile range from the box

manually counted the number of CD8 expressing cells located intratumorally vs. in the peritumoral stroma. As shown in Fig. 7, specimens with higher vs. lower EMT/Stroma_core gene expression exhibited significantly lower numbers of intratumoral CD8 cells (two-sided Wilcoxon rank sum test statistic = 25, $p = 0.035$) and a significantly higher ratio of stromal to intratumoral CD8 cells (two-sided Wilcoxon rank sum test statistic = 87, $p = 0.026$).

## Discussion

Immune checkpoint blockade, with anti-PD-1/PD-L1 antibodies, is now a standard treatment for platinum-resistant metastatic UC. However, only a subset of patients responds to treatment, highlighting the need to identify mechanisms of intrinsic resistance. Here, we have shown that while EMT-related gene expression and T-cell infiltration are positively correlated, the balance of these parameters may have prognostic/predictive implications in patients with advanced UC treated with PD-1 blockade.

The inverse correlation between tumor purity and EMT-related gene expression, strong positive correlation between EMT-related and stromal-related gene signatures and contribution of mouse reads to EMT-related gene expression in our UC PDX models all support the notion that stromal cells are a key source of EMT-related gene expression in UC. Our findings are consistent with recent studies in colorectal cancer and head and neck cancer

employing PDX models or single cell RNA-seq[31,32,36]. For example, using single cell RNA-seq data from 11 primary colorectal cancers and matched normal mucosa, Li et al. showed that EMT-related genes were found to be upregulated only in the cancer-associated fibroblast subpopulation of the tumor samples[32]. Still, stromal elements, such as cancer-associated fibroblasts, have been shown to induce the biological process of EMT in model systems and cancer-associated fibroblasts have been posited to even possibly arise from epithelial cancer cells undergoing EMT suggesting these cellular compartments and processes may be highly intertwined[37,38] complicating definitive dissection of each cellular compartment to EMT-related gene expression in the current study. Furthermore, partial EMT states may exist in only a subset of cancer cells, and in a dynamic fashion as shown in recent studies of other cancer types[36]. Future studies incorporating single cell RNA-seq in UC may shed further light on this subject.

We demonstrated that the balance of T-cell infiltration and EMT-related gene expression has potentially prognostic, and/or predictive, implications in both patients with clinically localized UC treated with cystectomy and in patients with advanced platinum-resistant UC treated with nivolumab, though the relationship between these variables differed slightly in the two cohorts (Supplementary Fig. 4). Specifically, in the TCGA cohort,

**Table 1 Impact of CD8 expression and EMT/Stroma_core gene expression on clinical outcomes in CheckMate 275 biomarker cohort ($n = 214$): $p$-values**

| Test | Model I | Model II | $p$-values* | | |
|---|---|---|---|---|---|
| | | | **PFS** | **OS** | **Objective Response** |
| Effect of CD8_IHC alone | Intercept | Model I + CD8_IHC | 1.20E−04 | 4.37E−03 | 1.20E−03 |
| Overall effect of adding EMT/Stroma_core | Intercept + CD8 IHC | Model I + EMT/Stroma_core + CD8_IHC:EMT/Stroma_core | 4.45E−02 | 3.49E−02 | 2.17E−02 |
| CD8_IHC:EMT/Stroma_core interaction | Intercept + CD8 IHC + EMT/Stroma_core | Model I + CD8_IHC:EMT/Stroma_core | 3.51E−02 | 3.96E−02 | 3.51E−02 |
| Effect of adding CD8_IHC to baseline variables | Intercept + HBN + LIVERMET + PDL1 | Model I + CD8_IHC | 1.04E−04 | 2.99E−03 | 3.57E−04 |
| Overall effect of adding EMT/Stroma_core to baseline variables and CD8_IHC | Intercept + HBN + LIVERMET + PDL1 + CD8 IHC | Model I + EMT/Stroma_core + CD8_IHC:EMT/Stroma_core | 5.73E−03 | 4.89E−03 | 1.67E−02 |
| CD8_IHC:EMT/Stroma_core interaction when baseline variables included | Intercept + HBN + LIVERMET + PDL1 + CD8_IHC + EMT/Stroma_core | Model I + CD8_IHC:EMT/Stroma_core | 9.00E−03 | 4.83E−03 | 3.07E−02 |
| Effect of EMT/Stroma_core alone | Intercept | Model I + EMT/Stroma_core | 6.53E−01 | 4.18E−01 | 4.60E−01 |
| Overall effect of adding CD8_IHC | Intercept + EMT/Stroma_core | Model I + CD8_IHC + CD8_IHC:EMT/Stroma_core | 3.01E−05 | 8.36E−04 | 1.50E−04 |
| Effect of adding EMT/Stroma_core to baseline variables | Intercept + HBN + LIVERMET + PDL1 | Model I + EMT/Stroma_core | 1.42E−01 | 2.12E−01 | 2.84E−01 |
| Overall effect of adding CD8_IHC to baseline variables and EMT/Stroma_Core | Intercept + HBN + LIVERMET + PDL1 + EMT/Stroma_core | Model I + CD8_IHC + CD8_IHC:EMTStroma_core | 9.03E−06 | 1.30E−04 | 5.04E−05 |

*$p$-values from likelihood-ratio hypothesis tests of the effects of CD8 IHC or EMT/Stroma_core scores on PFS, OS, and Objective Response, for the CheckMate 275 biomarker cohort. Tests for PFS and OS are from Cox PH models. Tests for objective response are from linear logistic regression models. Each test compares Model II to Model I. CD8_IHC, CD8 immunohistochemistry; EMT/Stromal_core, EMT/Stromal_core gene expression; HBN, hemoglobin; LIVERMET, presence of liver metastases; PDL1, PD-1 IHC score

ITA was positively correlated with OS and EMT-related gene expression was negatively correlated with OS and there was no significant statistical interaction between the parameters. In contrast, in the CheckMate 275 cohort, a significant statistical interaction was observed between these parameters; that is, the impact of EMT-related gene expression on objective response, PFS, and OS with nivolumab was observed only in patients with tumors harboring increased T-cell infiltration. There are practical and mechanistic reasons that might account for these differences Most importantly, the cohorts represent highly distinct clinical disease states associated with different treatments and prognoses. The assays used to measure T-cell infiltration and EMT-related gene expression also differed between the groups. Finally, the impact of EMT-related gene expression on prognosis in cystectomy-treated patients with localized disease could reflect additional biological processes (e.g., invasion, metastatic capacity, etc) beyond those related to immune modulation. On the other hand, the immunomodulatory effects might dominate the negative impact of EMT-related gene expression in patients with advanced disease treated with immune checkpoint blockade such that the effect is observed predominantly in tumors with increased T-cell infiltration. Nonetheless, the finding that the balance of these parameters confers potentially clinically relevant information in both disease settings is striking and further clinical validation and mechanistic exploration is warranted in both settings.

There are several possible underlying mechanisms that might contribute to the impact of EMT-related gene expression and T-cell infiltration on outcomes. Cancer-associated fibroblasts are among the most abundant cellular components of the stroma and contribute to tumor growth via secretion of pro-angiogenic signals and other growth factors, chemokines, and through orchestration of the composition of the extracellular maxtrix (ECM)[39]. Recently, using data from a large phase 2 study of patients with metastatic UC treated with the PD-L1 inhibitor

atezolizumab, Mariathasan et al demonstrated that a lack of response was associated with a signature of transforming growth factor-β signaling (TGF-β) derived from fibroblasts[40]. The investigators further demonstrated that increased TGF-β-signaling occurred particularly in tumors that showed exclusion of T-cells from the tumor parenchyma that were instead found in the peritumoral stroma. In our preliminary analysis of the spatial localization of T-cells in specimens with increased EMT-related gene expression, we made a similar observation. Intriguingly, Salmon et al, using immunostaining and real-time imaging of T-cells in viable slices of human lung tumors, showed that dense networks of fibronectin and collagen fibers surrounding the tumor bed were associated with reduced ability of T-cells to migrate and contact tumor cells[41]. Such findings may unify the observed stromal location of T-cells in tumors with increased EMT/Stroma and/or TGF-β expression and support exploration of strategies combining PD-1 blockade not only with therapies directed at stromal cells and stroma-related cytokines but also with therapies targeting the ECM directly.

There are potential limitations to our study. The single-arm nature of TCGA and CheckMate 275 cohorts limits the ability to dissect the prognostic vs. predictive nature of the T-cell and EMT-related gene expression balance. We did not use whole-genome transcriptomic data from the CheckMate 275 dataset but rather gene expression data derived from defined gene panels and therefore focused on the EMT/Stroma_core gene set for testing in this dataset, reasoning that this approach would best facilitate evaluation of the clinical application of the identified biomarkers in future studies. Finally, our analysis is retrospective in nature and validation of the findings in additional datasets is warranted.

Molecular subtypes of UC have been defined by several groups, including TCGA, and linked with prognosis and response to treatment. However, in prior analyses, the UC subtypes associated with the highest response rates to PD-1/PD-L1 blockade have been inconsistent and responses have been observed across all

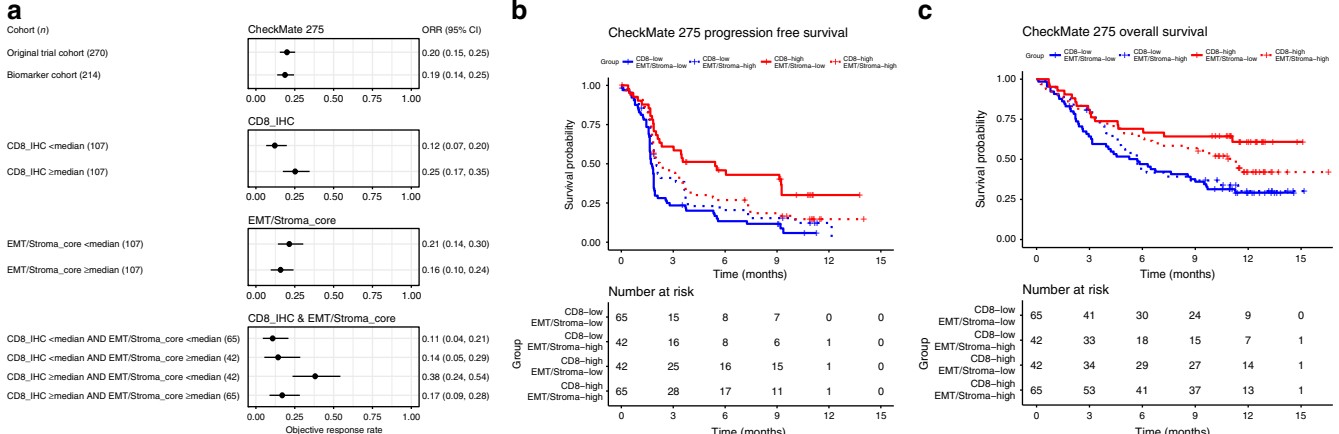

**Fig. 6** Higher EMT/Stroma-related gene expression is associated with an attenuated response to PD-1 blockade in T-cell infiltrated UC. **a** Objective response rate estimates with the PD-1 inhibitor, nivolumab, in the CheckMate 275 biomarker cohort, by CD8 and EMT/Stroma_core subgroup ($n = 214$). Subgroups are defined by biomarker score ≥ or < the median score. Plotting symbols show objective response rate point estimates; error bars show 95% confidence intervals for objective response rate. Kaplan–Meier estimates of **b** PFS and **c** OS curves in patients in the CheckMate 275 biomarker cohort, stratified according to the four biomarker subgroups

subtypes[2,6]. The luminal-infiltrated and basal-squamous TCGA UC subtypes have been proposed for prioritization for immune checkpoint blockade based on the relatively high infiltration of immune cells but these subtypes are also enriched in EMT-related genes providing potential insight for responses observed in only a subset of patients[30]. Focusing on established molecular subtypes of UC as a means of selecting patients for immune checkpoint blockade may overlook important biology that is relevant to sensitivity/resistance and shared across subtypes.

Here, we show that EMT-related gene expression and T-cell infiltration are positively correlated in UC yet confer disparate treatment response and prognostic information. EMT-related gene expression in UC, typically ascribed to the biological process of EMT in the epithelial compartment, may require reinterpretation given the key contribution of stromal cells to such gene expression. The balance of T-cell vs. EMT/stromal elements may provide a more informative snapshot of the antitumor immune response than measures of T-cells alone. Future work is focused on validation of these potential prognostic and/or predictive biomarkers, dissecting the mechanistic basis for our observations, and exploring regimens combining therapies targeting stromal elements plus PD-1/PD-L1 blockade in UC.

## Methods

**Analysis of TCGA bladder cancer dataset.** Bladder cancer RNAseq gene expression data ("Level_3_RSEM_genes_normalized") and patient survival data were downloaded from Firehose (2016_01_28) at the Broad Institute (https://confluence.broadinstitute.org/display/GDAC/Home/).

Estimation of ITA: To derive T-cell markers, we downloaded the gene expression profiles of 513 cell type markers across 22 different types and states of immune cells used by CIBERSORT[42]. Genes with standardized (gene-wise) expression value >2 in at least one T-cell subtype/status were considered T-cell markers resulting in a set of 144 T-cell markers. For each sample in the TCGA, ITA was estimated by the arithmetic mean of the 144 T-cell marker expression levels (in the log2 scale). We also used other sets of T-cell markers. The resultant ITA showed very strong correlation with each other (see Supplementary Note 1 and Supplementary Fig. 11).

EMT-related gene expression: The EMT-related gene expression signature was comprised of 200 genes obtained from gene set "hallmark epithelial mesenchymal transition" in The Molecular Signatures Database (MSigDB, software. broadinstitute.org/gsea/msigdb). For each sample in TCGA, an EMT-related gene expression score was calculated by the arithmetic mean of these 200 EMT gene expression levels (in the log2 scale). We also calculated EMT scores based on previous studies[15,26] and obtained similar results (see Supplementary Note 1)

Purity analysis: To account for purity in analyzing correlation between ITA and EMT-related genes, we downloaded tumor purity estimation for TCGA bladder samples from previous studies[23,43], We then adjusted the ITA value by the purity estimation. Specifically, we used a linear regression model $ITA \sim 1 + \log(1 - purity)$,

and obtained the residual of the model as the purity-adjusted ITA value. Similarly, we adjusted EMT-related gene expression by purity. We then calculated correlation between purity-adjusted ITA and EMT gene expression.

Survival analysis: Cox proportional hazards (PH) regression models were used to assess the dependence of overall survival on ITA and EMT-related gene expression. The magnitudes of associations were summarized by hazard ratios (HRs). Reported HRs were scaled to compare the 75th and 25th percentiles of the biomarker scores. Because the effects of the biomarkers were constrained to be linear in these models, the HR estimates depended only on the difference between the 75th and 25th percentiles, not on their individual values. Two-sided 95% confidence intervals for HRs were based on Wald test statistics. Kaplan–Meier plots based on categorization of the biomarker scores were used to illustrate associations with OS. To obtain the EMT/Stroma_core gene list, we assessed the association of each individual EMT gene with OS while controlling the effects of ITA using the bivariate Cox regression model, $\lambda(t|ITA, EMTgene_j) = \lambda_0(t)\exp(\beta_1 \times ITA + \beta_2 \times EMTgene_j)$. The EMT genes were then ranked according to the $p$-value by Wald test. We used a stringent cutoff of $p < 1e-6$ to select 18 EMT/Stroma_core genes (corresponding to adjusted $p < 2e-5$ after correction for multiple testing).

Multiple testing: The BH method[44] was used for correction of multiple testing when appropriate.

**Analysis of patient derived xenograft model.** We generated a cohort of 5 patient-derived xenograft (PDX) models from circulating tumor cells derived from the peripheral blood of patients with UC using an approach that we previously reported[45]. Briefly, 30 ml of peripheral blood was collected from patients with metastatic UC prior to initiation of chemotherapy. Density gradient centrifugation was performed followed by isolation of circulating tumor cells through flow cytometry by depletion of CD45+ mononuclear cells and subsequent subcutaneous injection of CD45- cells into immunocompromised "NSG" mice. PDX that had undergone less than 5 passages were histologically confirmed and molecularly characterized by performing genome-wide transcriptome profiling using RNA-sequencing. Reads were first mapped to human and mouse genome separately using TopHat[46]. The Bamcmp algorithm[34] was then used separate each read into human only, mouse only and both. The latter category was further categorized into reads aligning better to human or reads aligning better to mouse. As shown in supplementary Fig. 7, the fraction of reads mapped to both genome (human_better and mouse_better) is very small (<2%), suggesting very few reads with an ambiguous source. Reads from human_only and human_better were subsequently combined to represent all reads from human. Similarly, mouse_only and mouse_better categories were combined. FeatureCounts[47] was then used to calculate the read counts for each gene in human and mouse separately. After genes with very low read counts were removed (those with read counts less than five in both human and mouse for all 5 PDX models), a total of 14,018 genes were considered including 189 of the 200 hallmark_EMT genes and 128 of the 141 ESTIMATE_stromal genes. For each gene and each sample, the proportion of reads coming from mouse is calculated as read count from mouse for that gene divided by the total read count from either mouse or human for that gene. To derive species-specific RPM value, the raw count for each gene from human and mouse was scaled according to the species-specific library size (total read counts from human or mouse). Expression change in mouse vs human for each gene was then calculated by mouse-specific RPM value divided by human-specific RPM value for that gene.

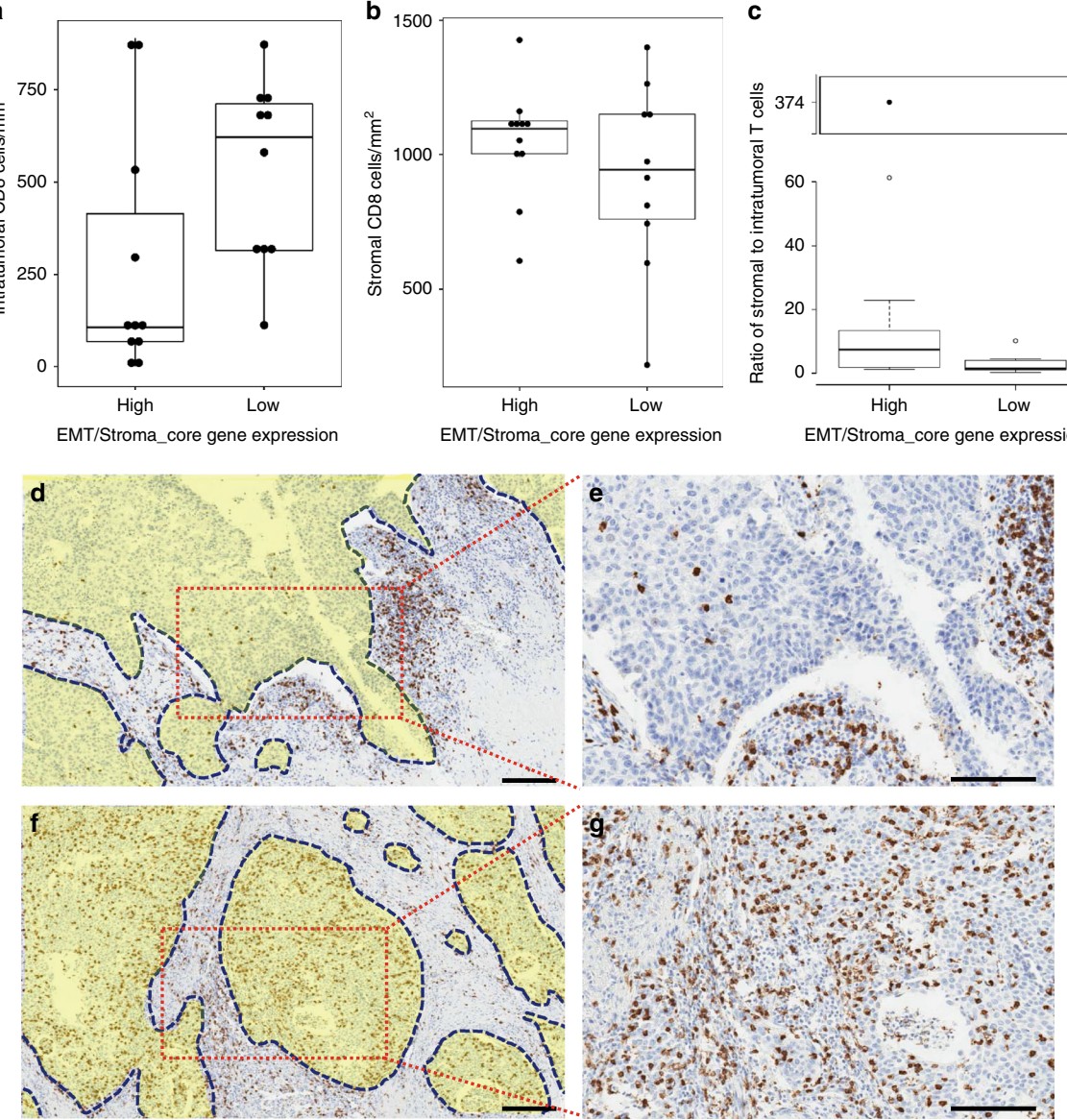

**Fig. 7** T-cells in UC specimens with higher EMT/Stroma-related gene expression are more frequently localized to the peritumoral stroma. Boxplots of the density of **a** intratumoral or **b** peritumoral-stroma CD8 cells in specimens (*n* = 21), by EMT/Stroma_core gene expression category (≥ or < median level); **c** Ratio of peritumoral stroma CD8 cells to intratumoral CD8 cells; **d**, **e** Representative tumor specimen with low intratumoral CD8 cells and high peritumoral stromal CD8 infiltrates, and **f**, **g** with high total and intratumoral CD8 cells. Yellow-shaded zones in images **d** and **f** identify tumor areas, whereas the rest represent adjacent stroma. Scale bars correspond to 200 μm. In boxplots, boxes extend from the first to third quartiles, middle line shows median, whiskers extend up to 1.5 times the interquartile range from the top and bottom of the box to the furthest data within that distance. Data beyond the end of the whiskers are "outlying" points. In **a** and **b**, filled circles show individual values. In **c**, the outlying point with extreme value is boxed with a break in the *Y* axis to better display the entire distribution of the data. Two lines at the end of the whiskers correspond to the maximum and minimum of the data points with the extreme outlier removed

**CheckMate 275 dataset and analysis**. The CheckMate 275 (NCT NCT02387996) dataset comprised 270 patients with platinum-resistant metastatic UC treated with nivolumab on a phase II clinical trial and has been previously described in detail[48]. Archival formalin-fixed paraffin embedded UC tumor specimens were submitted for each patient prior to initiation of nivolumab. The response assessments and survival follow-up of this cohort has previously been described; the objective responses were determined based on a blinded independent review committee assessment[48].

Gene expression was measured using the HTG EdgeSeq system (HTG Molecular, Tuczon, AZ) Oncology and Immuno-Oncology Biomarker Panels and has been previously described[48]. Data were transformed into log2 Trimmed mean of M-values (TMM) normalized counts per million (CPM) prior to analysis based on manufacturer's instructions. The EMT/Stroma_core score was calculated by the arithmetic mean of the 8 EMT/Stroma_core gene expression levels.

CD8 expression (mouse clone C8/144B, Dako North America, Carpenteria, CA, USA) was assessed by immunohistochemistry using an automated commercial

proprietary assay in a central laboratory (Mosaic laboratories, Lake Forest CA). Regions of interest were circled loosely around areas of tumor and CD8 infiltrates were expressed as % of total tumor area. Tumor cell PD-L1 membrane expression was assessed at a central laboratory (Dako PD-L1 immunohistochemical 28-8 pharmDx kit, Dako North America, Carpenteria, CA, USA).

Cox Proportional Hazards (PH) regression models were used to assess the dependence of PFS or OS on CD8 IHC score and stroma_core score. The models included linear effects of each biomarker and the multiplicative interaction between them. Proportional hazards assumptions were assessed by examination of scaled Schoenfeld residuals. For all Cox PH models, the PH assumption appeared reasonable. The magnitudes of associations were summarized by HRs, scaled as described above for TCGA dataset. Linear logistic regression models were used to assess the dependence of objective response on the biomarker scores. The magnitudes of associations were summarized by odds ratios (ORs), scaled in the same way as the reported HRs. Two-sided 95% confidence intervals for ORs were based on Wald test statistics. Two-sided 95% confidence intervals

for objective response rate were estimated by the Clopper-Pearson exact method[44].

Likelihood-ratio tests (two-sided) were used to test overall biomarker and interaction effects. No formal correction for multiple hypothesis testing was done. Kaplan–Meier plots based on categorization of the biomarker scores were used to illustrate associations with PFS or OS. All data analyses were performed with R 3.4.1 for Linux.

To assess whether the CD8 lymphocytic infiltrate was located intratumorally (mixed with the tumor cells) or in the peritumoral stroma, a subset of CD8 immunohistochemically stained slides from specimens with EMT/Stroma_core gene expression at or above the median expression level and from 21 specimens with EMT/Stroma_core gene expression below the median was selected. A genitourinary pathologist (M.C-M.), blinded to gene expression data, manually counted the number of CD8 expressing cells in five 200x microscopic fields for each of the histological compartments (intratumoral vs. peritumoral stroma), and then calculated the mean to determine the number of CD8 cells/mm$^2$ for each sample. The quantity of CD8 cells located in the different compartments in specimens with higher vs. lower EMT/Stroma_core gene expression was compared using the Wilcoxon rank sum test.

## Data availability

Raw FASTQ data and processed data for PDX model are publically available in the NCBI Gene Expression Omnibus (GSE116159). All other remaining data are available within the Article and Supplementary Files, or available from the authors upon request.

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

## Acknowledgements

The CheckMate 275 study was funded by Bristol-Myers Squibb. Biostatistical analysis was supported, in part, by P30 CA196521, The results shown here are in part based upon data generated by the TCGA Research Network: http://cancergenome.nih.gov/.

## Author contributions

L.W., A.S., P.M.S., J.Z. and M.D.G involved in discussion of the conception and design of the study. J.Z. and M.D.G. provided financial and administrative support. A.S., P.M.S., J. D., A.S.R, P.S., A.D., A.A. and M.D.G. provided the study materials. L.W., A.S., P.M.S., S. D.C., M.C.M, J.D., A.D. and M.D.G participated in data collection and analysis. L.W., A. S., P.M.S., S.D.C., M.C.M, J.D., Y.G., J.P.S., W.K.O., D.M., L.H., C.C, H.S., N.B., J.Z., P.S., A.S.R. and M.D.G contributed in manuscript writing. All authors gave final approval of the manuscript.Data availabilityRaw FASTQ data and processed data for PDX model are publically available in the NCBI Gene Expression Omnibus (GSE116159). All other remaining data are available within the Article and Supplementary Files, or available from the authors upon request.

## Additional information

**Competing interests:** M.D.G. has served as a consultant and has received research funding from Bristol-Myers Squibb. A.S., P.M.S., S.D.C., and A.A. are employees of Bristol-Myers Squibb. The remaining authors declare no competing interests.

