## [Peer Review File · Nature Communications]

Reviewers' comments:

Reviewer #1 (Remarks to the Author):

Summary: Overall an interesting study by Wang et colleagues who have investigated how three parameters (1-Epithelial-mesenchymal transition gene expression, 2-T cell infiltration and 3-disease progression) can be interconnected in the context of urothelial cancer.

- In the first part of the work, by extracting information from TCGA data with a focus on Bladder cancers, the authors confirmed an association between EMT-associated signature and increased immune contexture (Relevant ref. Mak MP Clin Can Res2016 ; PMID:26420858), focusing here on T cell infiltration (ITA).

- In a second part, authors have investigated the prognostic value of these biological parameters. This part is quite original in showing that EMT-related gene expression and T cell infiltration (ITA) have different prognostic values. Notably, EMT content in patients' tumors was associated with poorer prognosis in Bladder cancers while T cell infiltration (ITA) marked for good prognosis in terms of PFS and OS. This is also interesting because previous attempts from other groups to associate EMT to disease progression were not convincing/or conclusive. Moreover, the data suggest that combining these parameters (ITA:EMT) could improve prognostication of the disease, at least for early stage disease, considering that the TCGA collection mainly derived from early disease.

- Aside from these findings, authors also provide data in support of the notion that stromal cells (not the cancer cells) may be the major source of EMT-related gene expression. This is an interesting aspect, my opinion is that it is still requires some validation to strengthens this statement (see my comments below for more details).

- In the last part, in an attempt to better predict response to anti-PD1 in metastatic patients, authors have tested the potential utility to combine CD8 (as a surrogate for T cell Infiltration) and EMT status in a valuable cohort of bladder cancer patients who received anti-PD1 immunotherapy (Checkmate 275). It is clearly an interesting survey but my impression is that this part could be improved. Currently it is not as strong as the previous analysis in terms of evidence and statistics, in part due to a smaller cohort with only one arm, but not only. This perhaps also suffers from lack of clarity. Only one main figure is dedicated to this part (+ sup. Tables). Yet, the results deriving from this survey are presented as one of, if not as, the main finding of the manuscript (in the title, in the abstract and throughout the text). My feeling is that some 'clarification' and/or perhaps some 'toning down' of the conclusions on this aspect of the manuscript would be important. An expert biostatistician to include in the set of reviewers should be better placed to assess the relevance of the findings.

- An interesting study to consider as "bioinformatics/correlative/clinical and related statistics" research paper that could influence thinking in the field. To my opinion, clarification and additional analyses should be addressed prior to publication. For broad audience including, but not limited to, cancer specialists, clinicians, bioinformaticians. However, with a risk that some immunologists, biologists and pathologists be frustrated not seeing even one pathological section of representative human tumor cases, or from the studied Patient-derived xenografts (PDX), stained for T cell infiltration, a few EMT-core marker or simply by H&E. It would also be judicious for the authors to include representative cases also to help illustrate their statement that stromal component mainly accounts for EMT-gene expression in urothelial carcinomas.

Important comments:

1) There seem to be differences in prognosis values of the different parameters when comparing results from the two cohorts. I realize that this represents different clinical situations, but it is felt that the use of generic terms such as "outcomes" for different clinical settings is not always appropriate and may be misleading, or lead to overstatement in some cases; at least makes it difficult to conclude on the effect of each parameter or combinations in the TCGA vs the checkmate 275 setting. This

requires some clarification to avoid misperceptions.

A few examples where it could be improved:

a- L228-248: Statistics (pvalues) are almost absent in this part of manuscript concerning the checkmate 275 analysis, when it is largely used in the rest of the manuscript for the TCGA.

b- if I understood correctly, EMT alone had no prognostic value on survival in the checkmate 275 but rather a predictive value for response to anti-PD1, when it was clearly associated with adverse outcomes in the TCGA survey.

c- In the Checkmate 275, despite the univariate analysis and curves, it remains unclear to me if adding EMT information to CD8 in the CD8:PD1 significantly impacts the prognosis/ outcomes. Based on the hazard ratios, and p values of the supplementary tables, the effect of EMT appears minimal. Is it the case? Authors state L232, "... in patients with CD8 infiltrated tumors, higher versus lower core-EMT gene expression was associated with a lower likelihood of achieving an objective response and shorter PFS and OS (Tables S2&S4&S6)". Although the TCGA analysis seems strong, I would be more cautious in view of the hazard ratios and p-values presented for the CD8:core_EMT?

d- Moreover, for consistency, in Figure 6, authors should include the survival curves for each group (CD8 low, high, EMT low, high), like they actually did in Figure 4 for the TCGA cohort, and edit the text appropriately.

e- Authors could also include a bit more in their discussion with regards to why T cell infiltration (ITA status) alone had no real impact on prognostic value in TCGA cohort, while in the checkmate 275, it was beneficial in terms of response and survival anti-PD1 regimen setting.

f- Authors: L243, "The objective response rate with nivolumab was 16.9% in patients with CD8 highcore_EMThigh tumors versus 39.0% in patients with CD8highcore_EMThigh tumors (Figure 6A)". For clarity, authors should provide here the number of patients in each group.

g- L256, Authors state: "Here, we show that in T-cell infiltrated tumors, higher versus lower EMT-related gene expression is associated with worse outcomes". Authors should define which outcomes instead of using a generic name (Overall survival, progression-free survival, response to therapy...)

2) Authors state: L289 1) "Indeed, in an exploratory pan-cancer analysis utilizing TCGA data, we demonstrated a significant correlation between EMT-related gene expression and ITA among various tumor types", "a decrease in correlation conditional on tumor purity, and a significant prognostic impact of the EMT:ITA ratio".

-Although the signatures used were obviously different between studies, previous work such as the one of Mak and colleagues have made similar observations and merits to be more acknowledged throughout the manuscript.

3) With regards to the tumor purity and the involvement of stromal compartment in EMT-gene expression, it is an important and interesting observation to further consider. Some comparative studies using published signatures needs to be performed to better statute on this aspect. As it is, it gives the impression that the issue of EMT-related (stromalCell) gene expression vs EMT-related (CancerCell) gene expression is rather a matter of which EMT signature/or gene list was chosen to start the analysis.

a-To my knowledge the "Hallmark_epithelial mesenchymal transition" used from MSigDataBase is not specifically designed for cancer studies. Therefore the choice (or the confinement of the study) of the Hallmark_EMT signature is therefore questionable.

To support their message, it would be keen for the authors to assess well-known "cancer EMT" signatures specifically generated from cancer-related studies. I recommend the pan-EMT signature of Mak MP et al. (Clin Can Res2016 PMID:26420858) and the "cancer-specific EMT signature" and the "generic EMT signature" of Tan TZ et al. (EMBO molecular Medicine 2014, PMID: 25214461). Again it

will be important to perform such analyses to strengthen the statement that in general EMT-gene expression derived mostly from stromal cells. Notably, Tan TZ in their analysis compared tumor vs tumor cell lines from various cancer types including bladder cancer, and found that the EMT scoring of both cell lines vs tumors (using their signatures) highly correlate suggesting that in this setting stroma-related genes have a limited influence on the EMT scoring of tumors.

b- Other suggestions to help the authors without going into single cell sequencing, include comparative analysis of microdissected tumors, stroma vs epithelial areas, possibly by using published data already available.

c- To note, in their study, Tan TZ et al. (EMBO molecular Medicine 2014, PMID: 25214461) analyzed microdissected vs non-microdissected data from breast tumors and found very similar trend of high EMT content in the BCa Claudin-low group, suggesting that the stromal compartment had in this context minor effects on EMT-gene expression profiles in this context. In this line, the current study and results presented in Supp. Figure 1 shows that the bladder claudin-low had the highest expression of EMT genes and ITA. Would authors consider claudin-low as a more stromal tumor in the context of bladder cancer?

d- The study of PDX could have been interesting in this regard, but obviously suffers from lack of information from which molecular type were derived those PDX (luminal, basal, claudin-low ?), not forgetting that the microenvironment of a PDX is probably quite different from human tumor microenvironment.

4) A general comment for the authors, from a biological standpoint, it becomes more and more clear that in most tumors (except for rare instances or subtypes (such as Claudin Low group in breast cancer, or in some sarcomatoid neoplasms), we are dealing with a lot more than Two EMT classes, that complete EMT process per se, if it happens, is rare occurring in only a few cancer cells evading their primary sites, secondary sites, or circulating in the body. However more common events such as plasticity, dedifferentiation, adaptation processes may lead to more intermediate EMT states/phenotypes which may also have consequences on tumor progression (Nieto et al Cell 2016, PMID: 27368099). Therefore I am not sure that discriminating groups by low EMT vs high EMT groups will not be soon outdated. Perhaps one has to think about using another formulation such as Mesenchymal content/feature, instead of EMT (High, Intermediate and Low).

5) Regarding my point about intermediate EMT states, my main suggestion here would be to perform additional analyses considering more than two EMT classes to assess prognostic values (could be include as supplementary information), similar to what's been done by Lou YL recently for their lung Adenocarcinomas (PMID:26851185). This may be important for future consideration

Minor comments:

- L158: "Conditioning on EMT-related gene expression in a bivariate Cox regression model, most immune cell types were associated with better survival though T-cells and NK-cells were the most significantly associated with OS". It is interesting information and although this study focused on T cells, one should not forget that NK cells might also have important implications. This merits a few words in the discussion.

- L88-93: authors state "Figure1A shows 144 T-cell markers that ... representing different T-cell subsets, exhibit similar expression profiles across UC specimens suggesting that the different types of T-cells infiltrate into UC in a largely coordinated".

It seems not totally true, if we consider the Treg subset for example, which is interesting, and authors could comment on this.

- L306-310: "CAFs..... ECM has been demonstrated to impair the activity of both innate and adaptive immune cells in model systems related both to the physical properties of the ECM and as well as by serving as a reservoir for suppressive growth factors". Although I understand their point here, the authors should not completely exclude the possibility that Mesenchymal/like carcinoma cells could also contribute to some degrees to immune suppression and immune resistance as emerging data support this view, summarized recently (Terry S et al. Mol Oncol. 2017 PMID: 28614624)

- L329: "However, focusing on 'intrinsic' molecular subtypes of UC as a means of selecting patients for immune checkpoint blockade may overlook important biology that is relevant to sensitivity/resistance across subtypes"

Is there a possibility that their findings could actually add or improve the definition of UC molecular subtypes? To make it more to sensitivity/resistance? If the authors have an opinion of this, it would be nice to add here.

- L289: typo detected "biological relevance our findings)

Reviewer #2 (Remarks to the Author):

Galsky and colleagues study the correlation between EMT gene expression and T-cell infiltration. They provide evidence through meta-analysis of TCGA that while infiltrating T-cell abundance and EMT-related gene expression are correlated, they provide disparate prognostic indications. This finding is substantiated through EMT related gene-bulk expression in urothelial cancer. The authors demonstrate that in urothelial cancer treated with Nivo, higher EMT-related gene expression is associated with lower response rate. The finding is intriguing that it suggests targeting stromal elements.

The reviewer finds that the informatic analysis is consistent with the authors primary conclusions, though largely sets the basis for the hypothesis to be evaluated experimentally. The first major finding that the correlation between gene-expression and ITA is dependent on tumor purity, lends to many hypothesis given that low tumor content could be also a sample quality issue. The authors further study find that these patients have lower survivability.

The results on the patient derived PDX models are intriguing. The reviewer wished there was more detail on the bioinformatic analysis, but the detail that is present seems reasonably consistent with thorough analysis. Overall it's an intriguing article, and seems to have reached a good place for publication and communication among colleagues.

Reviewer #3 (Remarks to the Author):

This is a comprehensive and meaningful manuscript. It has significant strengths in the methods and the results are clinically relevant.

Issues to address:

There is discussion about hot and cold tumors, however no discussion on excluded tumors. There is the possibility the current classification underplays the relevance of this 3rd type.

A section on statistics is required. Multiple testing occurred and gene signatures were not predefined. Therefore the work is exploratory in nature.

The quadri low TCGA subtype is not well characterised. Currently the most recent update splits into 5

groups. There needs to be an initial investigation in the established subtypes if they are to be used at all. I don't think this part adds much. There is for example nothing in the discussion about this. The section on tumor purity is not well explained and requires clarification. There also needs a section in the discussion as to its relevance.

Figure 4b indicates a trend towards increased survival. There is multiple testing and no formal statistical plan. Therefore this statement is difficult to justify.

I think the discussion should be shortened. Parts of the discussion should move to the results. The methodology lack some detail. For example the purity experiments.

Reviewers' comments:

Reviewer #1

Summary: Overall an interesting study by Wang et colleagues who have investigated how three parameters (1-Epithelial-mesenchymal transition gene expression, 2-T cell infiltration and 3-disease progression) can be interconnected in the context of urothelial cancer.

- In the first part of the work, by extracting information from TCGA data with a focus on Bladder cancers, the authors confirmed an association between EMT-associated signature and increased immune contexture (Relevant ref. Mak MP Clin Can Res2016 ; PMID:26420858), focusing here on T cell infiltration (ITA).

- In a second part, authors have investigated the prognostic value of these biological parameters. This part is quite original in showing that EMT-related gene expression and T cell infiltration (ITA) have different prognostic values. Notably, EMT content in patients' tumors was associated with poorer prognosis in Bladder cancers while T cell infiltration (ITA) marked for good prognosis in terms of PFS and OS. This is also interesting because previous attempts from other groups to associate EMT to disease progression were not convincing/or conclusive. Moreover, the data suggest that combining these parameters (ITA:EMT) could improve prognostication of the disease, at least for early stage disease, considering that the TCGA collection mainly derived from early disease.

- Aside from these findings, authors also provide data in support of the notion that stromal cells (not the cancer cells) may be the major source of EMT-related gene expression. This is an interesting aspect, my opinion is that it still requires some validation to strengthens this statement (see my comments below for more details).

- In the last part, in an attempt to better predict response to anti-PD1 in metastatic patients, authors have tested the potential utility to combine CD8 (as a surrogate for T cell Infiltration) and EMT status in a valuable cohort of bladder cancer patients who received anti-PD1 immunotherapy (Checkmate 275). It is clearly an interesting survey but my impression is that this part could be improved. Currently it is not as strong as the previous analysis in terms of evidence and statistics, in part due to a smaller cohort with only one arm, but not only. This perhaps also suffers from lack of clarity. Only one main figure is dedicated to this part (+ sup. Tables). Yet, the results deriving from this survey are presented as one of, if not as, the main finding of the manuscript (in the title, in the abstract and throughout the text). My feeling is that some 'clarification' and/or perhaps some 'toning down' of the conclusions on this aspect of the manuscript would be important. An expert biostatistician to include in the set of reviewers should be better placed to assess the relevance of the findings.

- An interesting study to consider as "bioinformatics/correlative/clinical and related statistics" research paper that could influence thinking in the field. To my opinion, clarification and additional analyses should be addressed prior to publication. For broad audience including, but not limited to, cancer specialists, clinicians, bioinformaticians. However, with a risk that some immunologists, biologists and pathologists be frustrated not seeing even one pathological section of representative human tumor cases, or from the studied Patient-derived xenografts (PDX), stained for T cell infiltration, a few EMT-core

marker or simply by H&E. It would also be judicious for the authors to include representative cases also to help illustrate their statement that stromal component mainly accounts for EMT-gene expression in urothelial carcinomas.

We thank the Reviewer for these favorable comments and very constructive feedback. We have used the feedback above, as well as the individual points highlighted below, to revise the manuscript. We feel that the final manuscript has been further strengthened by these important suggestions.

We have taken several steps to address the Reviewer's comments including refining our analysis and presentation of the CheckMate 275 cohort data and providing better explanation and visualization of the results. In addition, we do feel that it is important to highlight that the goal of our analysis, unlike others that may be more commonly encountered in the literature, was not to establish a discovery and validation cohort to test a biomarker per se. Rather, we sought to define a process of potential immunobiological relevance in a cohort of patients who were not treated with immune checkpoint blockade and subsequently to test the potential therapeutic relevance of that process in a cohort of immune checkpoint blockade treated patients.

In summary,

- The manuscript has undergone thorough biostatistical review and revision.*
- We have revised the manuscript to minimize the use of terms such as “outcomes” and include more specific terminology regarding the particular outcome measures.*
- We have revised the figures to improve visualization of the results.*
- We have included a figure to enhance contextualization of the impact of T cell infiltration and EMT-related gene expression on survival in the two cohorts.*
- We have included an analysis exploring the spatial localization of T cells in specimens with higher versus lower EMT-related gene expression.*

Important comments:

1) There seem to be differences in prognosis values of the different parameters when comparing results from the two cohorts. I realize that this represents different clinical situations, but it is felt that the use of generic terms such as “outcomes” for different clinical settings is not always appropriate and may be misleading, or lead to overstatement in some cases; at least makes it difficult to conclude on the effect of each parameter or combinations in the TCGA vs the checkmate 275 setting. This requires some clarification to avoid misperceptions.

We thank the reviewer for this comment and for allowing us the opportunity to clarify. As the reviewer correctly pointed out, our cohorts differed in several important ways. The TCGA cohort was comprised of 408 patients with clinically localized urothelial cancer of the bladder treated with cystectomy. RNA sequencing data from this cohort was utilized to derive both measures of T cell infiltration and EMT-related gene expression. The CheckMate 275 cohort was comprised of 214 patients with platinum-resistant metastatic

disease treated with nivolumab. T cell infiltration was measured using immunohistochemistry for CD8 and EMT-related gene expression was measured using a targeted gene expression profiling platform.

In both cohorts, we demonstrated that the balance of T cell infiltration and EMT-related gene expression has potential prognostic (and/or predictive in CheckMate cohort) implications, though the relationship between these variables differed slightly in the two cohorts. Specifically, in the TCGA cohort, T cell infiltration was positively correlated with overall survival and EMT-related gene expression was negatively correlated with overall survival and there was no significant statistical interaction between the parameters. In contrast, in the CheckMate 275 cohort, a significant statistical interaction was observed between these parameters; that is, the negative impact of EMT-related gene expression on response, progression-free survival, and overall survival with nivolumab was observed only in patients with tumors harboring higher T cell infiltration. The manuscript has been revised to clarify these findings and the following figure has been included to provide a clear visualization of the relationship between the parameters and outcomes in the two cohorts. In addition, we have expanded the discussion to highlight both potential practical, and mechanistic, explanations for these findings.

Not only are there potential practical explanations for these observations (i.e., differences in disease states and assays) but there are also potential mechanistic explanations. For example, the impact of EMT-related gene expression on prognosis in cystectomy-treated patients with localized disease could reflect additional biological processes (e.g., invasion, metastatic capacity, etc) beyond those related to immune modulation. On the other hand, the immunomodulatory effects might dominate the negative impact of EMT-related gene expression in patients with advanced disease treated with immune checkpoint blockade such that the effect is observed predominantly in tumors with increased T cell infiltration.

A few examples where it could be improved:

a- L228-248: Statistics (pvalues) are almost absent in this part of manuscript concerning the checkmate 275 analysis, when it is largely used in the rest of the manuscript for the TCGA.

We have addressed this concern in the revised manuscript.

b- If I understood correctly, EMT alone had no prognostic value on survival in the checkmate 275 but rather a predictive value for response to anti-PD1, when it was clearly associated with adverse outcomes in the TCGA survey.

Please see our response above for clarification and Table 1, Figure 7, and Figures S8-11 in the revised manuscript.

c- In the Checkmate 275, despite the univariate analysis and curves, it remains unclear to me if adding EMT information to CD8 in the CD8:PD1 significantly impacts the prognosis/outcomes. Based on the hazard ratios, and p values of the supplementary tables, the effect of EMT appears minimal. Is it the case? Authors state L232, "... in patients with CD8 infiltrated tumors, higher versus lower core-EMT gene expression was associated with a lower likelihood of achieving an objective response and shorter PFS and OS (Tables S2&S4&S6)". Although the TCGA analysis seems strong, I would be more cautious in view of the hazard ratios and p-values presented for the CD8:core_EMT ?

Please see our response above for clarification and Table 1, Figure 7, and Figures S8-11 in the revised manuscript.

d- Moreover, for consistency, in Figure 6, authors should include the survival curves for each group (CD8 low, high, EMT low, high), like they actually did in Figure 4 for the TCGA cohort, and edit the text appropriately.

We feel that Figure S9 provides better visualization of the relationship between the two parameters (as continuous variables) in TCGA and the CheckMate 275 cohorts than dichotomizing the variables in the survival curves. However, we have also now included the requested survival curves with the variables dichotomized in Figure S8.

e- Authors could also include a bit more in their discussion with regards to why T cell infiltration (ITA status) alone had no real impact on prognostic value in TCGA cohort, while in the checkmate 275, it was beneficial in terms of response and survival anti-PD1 regimen setting.

This is an important observation. The Reviewer correctly highlights the finding that as a continuous variable, our gene expression based measure of T cell infiltration (ITA) was not significantly correlated with survival on univariate analysis in the TCGA cohort while CD8 expression as measured by immunohistochemistry was significantly correlated with survival on univariate analysis in the CheckMate 275 cohort. As highlighted above, we believe this is likely related to both the difference in the disease states/treatment between the two cohorts in addition to the different approaches to measure T cell infiltration (gene expression versus

immunohistochemistry). Mechanistically, one might expect to see a stronger prognostic (or potentially predictive) impact of baseline T cell infiltration in an immune checkpoint inhibitor-treated population. However, one might also (correctly) point out that tumor infiltrating lymphocytes (as measured by immunohistochemistry) have previously been shown to be associated with survival in patients with muscle-invasive bladder cancer treated with cystectomy alone. Importantly, in the most frequently cited study supporting the relationship between tumor infiltrating lymphocytes (TILs) and survival in patients with muscle-invasive urothelial cancer treated with cystectomy (Sharma et al, PNAS, 2007), only intratumoral TILs were considered because the authors prior work “demonstrated that intratumoral TILs, as opposed to stromal TILs, correlated with favorable clinical outcomes”. Gene expression based measures of T cell infiltration alone do not impart information regarding spatial localization of T cells which may explain the lack of correlation between ITA alone and survival in TCGA. We have expanded on this notion in the revised Results and Discussion and we feel that the prognostic/predictive information derived from the combined measures of T cell infiltration and EMT-related gene expression may actually be related to, at least in part, imparting information related to T cell localization (i.e, intratumoral T cells versus T cell confined to the stroma).

f- Authors: L243, “The objective response rate with nivolumab was 16.9% in patients with CD8 highcore_EMThigh tumors versus 39.0% in patients with CD8highcore_EMFlow tumors (Figure 6A)”. For clarity, authors should provide here the number of patients in each group.

To enhance visualization and interpretation of response rates in these subgroups, we have now included Figure 6A which includes response rates and 95% confidence intervals, in addition to the numbers of patients, for each subgroup.

g- L256, Authors state: “Here, we show that in T-cell infiltrated tumors, higher versus lower EMT-related gene expression is associated with worse outcomes”. Authors should define which outcomes instead of using a generic name (Overall survival, progression-free survival, response to therapy...)

We have included the specific outcome measures where appropriate throughout the manuscript.

2) Authors state: L289 1) “Indeed, in an exploratory pan-cancer analysis utilizing TCGA data, we demonstrated a significant correlation between EMT-related gene expression and ITA among various tumor types”, ”a decrease in correlation conditional on tumor purity, and a significant prognostic impact of the EMT:ITA ratio”.

-Although the signatures used were obviously different between studies, previous work such as the one of Mak and colleagues have made similar observations and merits to be more acknowledged throughout the manuscript.

We have added additional reference to the work by Mak and colleagues and have also included the EMT-related signatures from Mak and colleagues in the revised manuscript as outlined below.

3) With regards to the tumor purity and the involvement of stromal compartment in EMT-gene expression, it is an important and interesting observation to further consider. Some comparative studies using published signatures needs to be performed to better statute on this aspect. As it is, it gives the impression that the issue of EMT-related (stromalCell) gene expression vs EMT-related (CancerCell) gene expression is rather a matter of which EMT signature/or gene list was chosen to start the analysis.

a-To my knowledge the “Hallmark_epithelial mesenchymal transition” used from MSigDataBase is not specifically designed for cancer studies. Therefore the choice (or the confinement of the study) of the Hallmark_EMT signature is therefore questionable.

To support their message, it would be keen for the authors to assess well-known “cancer EMT” signatures specifically generated from cancer-related studies. I recommend the pan-EMT signature of Mak MP et al. (Clin Can Res 2016 PMID:26420858) and the “cancer-specific EMT signature” and the “generic EMT signature” of Tan TZ et al. (EMBO molecular Medicine 2014, PMID: 25214461). Again it will be important to perform such analyses to strengthen the statement that in general EMT-gene expression derived mostly from stromal cells.

Notably, Tan TZ in their analysis compared tumor vs tumor cell lines from various cancer types including bladder cancer, and found that the EMT scoring of both cell lines vs tumors (using their signatures) highly correlate suggesting that in this setting stroma-related genes have a limited influence on the EMT scoring of tumors.

b- Other suggestions to help the authors without going into single cell sequencing, include comparative analysis of microdissected tumors, stroma vs epithelial areas, possibly by using published data already available.

c- To note, in their study, Tan TZ et al. (EMBO molecular Medicine 2014, PMID: 25214461) analyzed microdissected vs non-microdissected data from breast tumors and found very similar trend of high EMT content in the BCa Claudin-low group, suggesting that the stromal compartment had in this context minor effects on EMT-gene expression profiles in this context. In this line, the current study and results presented in Supp. Figure 1 shows that the bladder claudin-low had the highest expression of EMT genes and ITA. Would authors consider claudin-low as a more stromal tumor in the context of bladder cancer?

d- The study of PDX could have been interesting in this regard, but obviously suffers from lack of information from which molecular type were derived those PDX (luminal, basal, claudin-low ?), not forgetting that the microenvironment of a PDX is probably quite different from human tumor microenvironment.

We very much appreciate the Reviewer's comments and we have responded to each comment in more detail below. However, first we would like to provide some additional context regarding the emerging literature relevant to this topic. Since most EMT studies have focused on laboratory models, the extent and significance of EMT in primary tumors and metastases remain controversial. Furthermore, while mesenchymal subtypes have been identified for certain tumors based on gene expression data, it remains unclear whether they reflect mesenchymal cancer cells or alternatively contributions of non-malignant mesenchymal cell types in the tumor microenvironment. There is a growing body of literature in other cancer types utilizing novel, robust, and complementary approaches supporting that indeed EMT-related gene expression is derived from stromal cells rather than cancer cells. Li et al (Nature Genetics 49, 708–718, 2017) performed an unbiased analysis of transcriptional heterogeneity in colorectal tumors and their microenvironments using single-cell RNA-seq from colorectal tumors and showed that epithelial–mesenchymal transition (EMT)-related genes were found to be upregulated only in the cancer associated fibroblast subpopulation of tumor samples. Remarkably, a recent analysis from Puram et al (Cell, 171, 1611-1624, 2017) utilizing single cell RNA sequencing of squamous cell carcinoma of the oral cavity revealed highly similar findings. Isella et al (Nature Genetics, 47, 312–319, 2015), utilizing colon cancer PDX models to distinguish the epithelial versus stromal contribution of gene expression, demonstrated that EMT-related genes were derived from the stromal compartment. Though we acknowledge that additional work is needed to identify the cellular source of EMT-related genes in urothelial cancer, and to better define the cross-talk between the stromal and epithelial compartments in situ, we feel that the comparisons with cell lines and the micro- versus macro-dissected samples are problematic in advancing our understanding of this issue. Rather, single cell RNA sequencing, and advanced computational approaches, are likely required to generate further insights and this work is being actively pursued by our group. We provide a more detailed response to the Reviewer's individual suggestions below.

a. The Hallmark_EMT gene set is commonly utilized in cancer studies. However, the Reviewer raises an important consideration and to address this, we have also evaluated the Mak Pan_EMT signature, the Tan Pan_EMT signature, and the Tan Bladder_Cancer_EMT signature. Interestingly, despite very few genes overlapping in these signatures, the EMT scores calculated from them are highly correlated. (Figures attached below). The following paragraph, together with the additional Figures, was in the Supplementary Results section of the revised manuscript.

“To assess how different sets of T-cell markers could affect the ITA estimation, we downloaded four EMT signature lists from the two previous studies: pan-EMT signature by Mak MP et al² (referred to as EMT_pancancer_Mak), generic-EMT signature by Tan et al³ (referred to as EMT_generic_Tan), bladder cancer-specific EMT signature by Tan et al³ (referred to as EMT_BLCA_Tan) and EMT signature derived from cancer cell lines by Tan et al³ (referred to as EMT_cellline_Tan). Note that the full BLCA-specific EMT list is not available from Tan's paper³, hence we used a subset of generic-EMT signature genes which the author annotated as belonging to BLCA-specific list. In contrast to the hallmark_EMT genes set which contain mostly mesenchymal markers, these four EMT gene lists contain both mesenchymal and epithelial markers. The EMT

score was calculated by mean expression difference between mesenchymal and epithelial markers, as described in the study of Mak MP et al². As shown in Figure S1, the EMT scores obtained using different EMT gene lists highly correlate with each other (Spearman's correlation >0.8). In addition, the relationship between these different EMT scores and ITA and between the different EMT scores and purity, were highly consistent with our results using the Hallmark_EMT gene set (data not shown). Thus, our observations were unlikely biased by the use of a particular EMT gene set.”

Overlap among “EMT_Hallmark” and mesenchymal markers (A)(B) and epithelial markers (C)(D) from other EMT signature gene lists.

Pair-wise correlation of EMT scores calculated based on different EMT gene lists. Spearman's correlation coefficient was shown in the upper right pattern.

Tan and colleagues computed EMT scores of bladder cancer cell lines based on gene expression data of BLA-40 and CCLE. They then utilized published data regarding these cell lines describing immunofluorescence staining of EMT markers, cell line morphology, and invasion assays based solely on the cell line name from published literature. In their Figure S1, they do show some relationship between a more “mesenchymal” EMT score and these “EMT features” though there were only 7 cell lines that scored more “mesenchymal” and at least one of these 7 cell lines had more epithelial features by morphology and IF. We agree that this is an interesting study that indicates that cultured cells that have more mesenchymal features are associated with a higher EMT gene score. However, this does not indicate that cancer cells in situ are a predominant source of this EMT-related expression in bulk transcriptomes and the artificial conditions of cancer cells grown in culture complicate such an extrapolation.

In response to the Reviewer's suggestion, for the EMT signature genes derived from cell lines (EMT_cellline_Tan), we estimated their expression fold change in stroma vs epithelial using our PDX models as well as the microdissected tumors (please refer to the next section for details of the microdissected datasets). As shown in below, the majority of these cell line mesenchymal markers have higher expression in stroma as compared to epithelial cells, while the epithelial markers have higher expression in epithelial cells as compared to stroma. In addition, the estimated log2FC for the cell line mesenchymal markers is similar to that of the Hallmark_EMT genes (rank test p-value>0.05). Thus, stroma still appears to

be a key source for those mesenchymal signature genes despite being derived from cancer cell lines.

Comparative analysis of gene expression in stroma vs epithelial area for EMT signature genes in EMT_cellline_Tan.

b. We were unable to locate a publically available microdissected data with stroma vs epithelial pairs in bladder cancer. However, we did find a few such datasets in colorectal, ovary and breast cancer. Similar to the analysis in PDX model, we compared the gene expression in epithelial vs stroma for EMT and stromal markers. Particularly, we obtained the median of the log2FC of each gene in stroma area vs the paired epithelial area. We plotted the log2FC estimated in each microdissected dataset against that in the PDX model, and observed significant correlation between them (shown below). Most of the stromal genes (>98% for all four datasets) and the majority of EMT genes (>83% for all four datasets) showed higher expression in stroma vs epithelial (log2FC>0). Thus, comparative analysis of microdissected tumors in various cancer types concurs that EMT genes have higher expression in stroma vs epithelial areas.

These findings further support that stromal cells serve as a key source for EMT-related gene expression.

Comparative analysis of gene expression in stroma vs epithelial area. (Colorectal cancer-GSE35602: 17 stroma-epithelial pairs of colorectal tumors; Breast cancer-DCIS-GSE14548: 9 stroma-epithelial pairs of breast tumors; Breast cancer-IDC-GSE14548: 9 stroma-epithelial pairs of breast tumors; Ovary cancer-GSE9890: 5 stroma-epithelial pairs of ovary tumor)

c. In the study from Tan and colleagues, in the laser capture microdissected (LCM) cohort, there was only one tumor classified as Claudin-low and only one tumor classified as Normal-like (the subtypes that were more mesenchymal in the non-LCM cohort). The vast majority of the samples in the LCM cohort scored as more epithelial using the Generic EMT Score in stark contrast to the non-LCM (Tan manuscript Figure 2 as shown below). Indeed, the finding that the Claudin-low and Normal-like subtypes barely existed in the LCM cohort suggests that LCM “removes” these subtypes and coupled with the overall “down-scoring” in the LCM cohort is supportive of our contention that the stroma contributes largely to EMT-related gene expression.

Non-Laser-Capture-Micro-dissected Cohort

Subtype	Basal	Claudin-Low	Luminal-A	Luminal-B	ERBB2+	Normal-like
p - value	2.0E-40	2.5E-68	0.0496	3.3E-79	2.5E-6	6.2E-54

Laser-Capture-Micro-dissected Cohort

Subtype	Basal	Claudin-Low	Luminal-A	Luminal-B	ERBB2+	Normal-like
p - value	1.1E-20	N.A	0.006	0.089	0.001	N.A

We do believe that claudin-low bladder cancer is likely more “stromal” as shown below.

Tumor purity (left) and EMT score after adjusting for purity (right) for three subtypes.

However, work regarding the molecular subtypes of bladder cancer has been progressing so rapidly that since our initial submission, the second TCGA bladder cancer subtyping classification was published. In response to comments from Reviewer 3, we have revised our manuscript to include the 5 TCGA molecular subtypes (luminal, luminal-papillary, luminal-infiltrated, basal-squamous, and neuronal subtypes) rather than subtyping based on claudin low, basal, and luminal subgroups.

d. We agree that the microenvironment of a PDX is different from that of a human tumor microenvironment. However, utilizing human versus mouse reads derived from PDX models as a means of interrogating tumor versus stroma has recently been utilized by several groups. As already noted above, Isella et al (Nature Genetics, 47, 312–319, 2015), utilized colon cancer PDX models to distinguish the epithelial versus stromal contribution of gene expression and similar to our analysis demonstrated that EMT-related genes were derived from the stromal compartment. Nicolle et al (Cell Reports, 21, 2458-2470, 2017) utilized pancreatic cancer PDX models and demonstrated that such models could be utilized to uncover the complex interplay between pancreatic cancer cells and the stroma leading to the identification of a novel therapeutic target involved in tumor-stromal cross-talk.

4) A general comment for the authors, from a biological standpoint, it becomes more and more clear that in most tumors (except for rare instances or subtypes (such as Claudin Low group in breast cancer, or in some sarcomatoid neoplasms), we are dealing with a lot more than Two EMT classes, that complete EMT process per se, if it happens, is rare occurring in only a few cancer cells evading their primary sites, secondary sites, or circulating in the body. However more common events such as plasticity, dedifferentiation, adaptation processes may lead to more intermediate EMT states/phenotypes which may also have consequences on tumor progression (Nieto et al Cell 2016, PMID: 27368099). Therefore I am not sure that discriminating groups by low EMT vs high EMT groups will not be soon outdated.

We completely agree that the identification of potential biological processes representing true epithelial-mesenchymal transition in human solid tumors is rapidly evolving and that this is likely both a dynamic process and not an all-or-none phenomenon. Particularly intriguing in this regard is the recent study from Puram and colleagues (Single-Cell Transcriptomic Analysis of Primary and Metastatic Tumor Ecosystems in Head and Neck Cancer, Cell, 2017). Puram and colleagues investigated primary head and neck tumors and matched lymph nodes using single cell RNA sequencing. These investigators identified a subset of cells that retained epithelial markers and also expressed a “partial EMT” program with extracellular matrix genes but lacking classical EMT transcription factors. Importantly, these “partial EMT” cells localized to the leading edge of primary tumors in proximity to cancer associated fibroblasts. Incidentally, these investigators also demonstrated that the mesenchymal TCGA subtype of head and neck cancer reflected high stromal (cancer associated fibroblast and monocyte) representation in bulk samples, rather than a distinct malignant cell program. This study nicely highlights the possibility of intermediate EMT states but also demonstrates the added complexity of cross talk between epithelial cells at the tumors leading edge and stromal elements.

It is important to point out that the goal of our study was to begin to dissect the basis, and potential clinical relevance, for the observation that existing EMT-related gene expression signatures and measures of T cell infiltrated are highly correlated rather than study the biological process of EMT per se. At the Reviewer's suggestion, we did perform an analysis similar to what has been done by Lou et al (see below).

5) Regarding my point about intermediate EMT states, my main suggestion here would be to perform additional analyses considering more than two EMT classes to assess prognostic values (could be include as supplementary information), similar to what's been done by Lou YL recently for their lung Adenocarcinomas (PMID:26851185). This may be important for future consideration

Lou et al utilized the following approach in their study, "For each sample, an EMT score was computed using an averaging scheme based on the mRNA expression of 76 genes previously published by our group and originally derived from analysis of NSCLC. The scores were calculated as the average expression level of "mesenchymal" genes minus the average expression level of "epithelial" genes. Tumor samples were then classified by EMT score as EMT-low (defined by EMT scores \leq lowest 1/3) or EMT-high (defined by EMT scores \geq highest 1/3) in both TCGA and PROSPECT datasets. By comparing the tumor samples with either relatively high or low EMT score, but not intermediate, we expected to increase the likelihood of identifying the immune markers associated with either "epithelial" or "mesenchymal" lung adenocarcinomas"

At the Reviewer's suggestion, we have used the same strategy as that used by Lou et al to generate more than two "EMT states" and demonstrated below.

Kaplan-Meier survival curves for UC patients in TCGA divided into three groups by the EMT-related gene expression (cut at tertiles).

Minor comments:

- L158: “Conditioning on EMT-related gene expression in a bivariate Cox regression model, most immune cell types were associated with better survival though T-cells and NK-cells were the most significantly associated with OS”. It is interesting information and although this study focused on T cells, one should not forget that NK cells might also have important implications. This merits a few words in the discussion.

We have included an additional statement about the immune cell types in the discussion.

- L88-93: authors state “Figure1A shows 144 T-cell markers that ... representing different T-cell subsets, exhibit similar expression profiles across UC specimens suggesting that the different types of T-cells infiltrate into UC in a largely coordinated”.

It seems not totally true, if we consider the Treg subset for example, which is interesting, and authors could comment on this.

We agree that there is some variability between immune cell types in different specimens and removed this statement from the manuscript to avoid confusion.

- L306-310: “CAFs..... ECM has been demonstrated to impair the activity of both innate and adaptive immune cells in model systems related both to the physical properties of the ECM and as well as by serving as a reservoir for suppressive growth factors”. Although I understand their point here, the authors should not completely exclude the possibility that Mesenchymal/like carcinoma cells could also contribute to some degrees to immune suppression and immune resistance as emerging data support this view, summarized recently (Terry S et al. Mol Oncol. 2017 PMID: 28614624)

We agree with this point and have included this in the discussion.

- L329: “However, focusing on ‘intrinsic’ molecular subtypes of UC as a means of selecting patients for immune checkpoint blockade may overlook important biology that is relevant to sensitivity/resistance across subtypes” Is there a possibility that their findings could actually add or improve the definition of UC molecular subtypes? To make it more to sensitivity/resistance? If the authors have an opinion of this, it would be nice to add here.

We have added the following section to the discussion:

Molecular subtypes of UC have been defined by several groups, including TCGA, and linked with prognosis and response to treatment. However, in prior analyses, the UC subtypes associated with the highest response rates to PD-1/PD-L1 blockade have been inconsistent and responses have been observed across all subtypes. The luminal-infiltrated and basal-squamous TCGA UC subtypes have been proposed for prioritization for immune checkpoint blockade based on the relatively high infiltration of immune cells but these subtypes are also enriched in EMT-related genes providing potential insight for responses observed in only a subset of patients with these subtypes. Focusing on established molecular subtypes of UC as

a means of selecting patients for immune checkpoint blockade may overlook important biology that is relevant to sensitivity/resistance and shared across subtypes.

- L289: typo detected (“biological relevance our findings)

We have deleted this sentence.

Reviewer #2

Galsky and colleagues study the correlation between EMT gene expression and T-cell infiltration. They provide evidence through meta-analysis of TCGA that while infiltrating T-cell abundance and EMT-related gene expression are correlated, they provide disparate prognostic indications. This finding is substantiated through EMT related gene-bulk expression in urothelial cancer. The authors demonstrate that in urothelial cancer treated with Nivo, higher EMT-related gene expression is associated with lower response rate. The finding is intriguing that it suggests targeting stromal elements.

The reviewer finds that the informatic analysis is consistent with the authors' primary conclusions, though largely sets the basis for the hypothesis to be evaluated experimentally. The first major finding that the correlation between gene-expression and ITA is dependent on tumor purity, lends to many hypotheses given that low tumor content could be also a sample quality issue. The authors' further study find that these patients have lower survivability.

The results on the patient derived PDX models are intriguing. The reviewer wished there was more detail on the bioinformatic analysis, but the detail that is present seems reasonably consistent with thorough analysis. Overall it's an intriguing article, and seems to have reached a good place for publication and communication among colleagues.

We thank the reviewer for these highly favorable comments. We have expanded upon the detail of the bioinformatics analysis. We have added additional details to the method section regarding survival analysis, purity analysis, PDX data analysis as well as data processing.

Reviewer #3

This is a comprehensive and meaningful manuscript. It has significant strengths in the methods and the results are clinically relevant.

We thank the reviewer for these highly favorable comments.

Issues to address:

There is discussion about hot and cold tumors, however no discussion on excluded tumors. There is the possibility the current classification underplays the relevance of this 3rd type.

The reviewer highlights a critical issue which we feel, in fact, may be a key finding from our analysis. Indeed, gene expression-based measures of T cell infiltration are increasingly being used in immunogenomic studies. These gene expression measures are used to distinguish “hot” from “cold” tumors but do not provide information regarding the spatial distribution of T cells within tumor specimens. Indeed, we believe that prognostic information derived from the balance of T cell infiltration and EMT-related gene expression in our study may actually be related to this very concept. That is, we believe that tumors with higher measures of T cell infiltration and increased EMT-related gene expression are likely the tumors with the “excluded” phenotype whereas tumors with higher measures of T cell infiltration and lower EMT-related gene expression are those tumors with intratumoral T cells. We have performed preliminary work supporting this hypothesis and this has been included in the revised manuscript.

A section on statistics is required. Multiple testing occurred and gene signatures were not predefined. Therefore the work is exploratory in nature.

The manuscript has undergone a thorough biostatistical review and revision. We corrected for multiple testing and reported adjusted p-value when applied. We have expanded on the Methods to provide much more detail regarding the analysis.

Please note that the original gene signature (Hallmark_EMT) was indeed predefined. The reviewer is correct that we did not correct for multiple testing in defining the 18 EMT/Stroma_core genes but rather these were selected based on a very stringent cutoff of p-value < 1e-6. We have thus added adjusted p-value and more details on how the 18 EMT/Stroma_core genes were selected in the method section: “To obtain the EMT/stroma_core gene list, we assessed the association of each individual EMT gene with OS while controlling the effects of ITA using the bivariate Cox regression model $\text{Surv}(\text{time}, \text{event}) \sim \text{ITA} + \text{EMTgene}_i$. The EMT genes were then ranked according to the p-value by Wald test. We chose a very stringent cutoff of $p < 1e-6$ to select 18 EMT/stroma_core genes (corresponding to adjusted $p < 2e-5$ after correction for multiple testing).” Still, we agree that the work is exploratory in nature and requires validation.

The claudin low TCGA subtype is not well characterized. Currently the most recent update splits into 5 groups. There needs to be an initial investigation in the established subtypes if they are to be used at all. I don't think this part adds much. There is for example nothing in the discussion about this.

We replaced the old subtyping with the most recently published TCGA subtyping of 5 groups: basal, luminal, luminal-infiltrated, luminal-papillary and neuronal. The figures below (Figure 2C and Figure S2 in the manuscript) show the boxplot of ITA and EMT in each subtype and demonstrate the correlation between EMT and ITA score for all TCGA samples as well as in each subtype. Although the correlation was partially driven by the covariation of ITA and EMT across different subtypes, it remained significant within some subtypes.

Boxplot of EMT(A) and ITA(B) across different subtypes of UC in TCGA.

Correlation between ITA and EMT in TCGA UC samples and in each subtype.

The section on tumor purity is not well explained and requires clarification. There also needs a section in the discussion as to its relevance.

We have revised the manuscript to clarify this section, adding detail to the methods, and have also included additional verbiage in the discussion regarding this topic. While from a

computational biology standpoint, we acknowledge that this section is somewhat complicated, the key messages from these analyses are relatively straightforward and as follows:

- *Bulk transcriptomes are comprised of genes derived from tumor cells and also genes derived from stroma, immune cells, and other elements.*
- *Samples with lower tumor purity have less cancer cells and more stromal and/or immune cells. While tumor purity is often interpreted as an indicator of sample quality, tumor purity may more importantly reflect multi-directional cross talk with the microenvironment. Therefore, tumors may be low purity not because the samples are poor quality but rather because the tumor and microenvironment interact in such a way to drive a robust immune and/or stromal reaction.*
- *Computational approaches can identify the tumor purity of a sample by “correcting” for the presence of a stromal and immune gene signature.*
- *The Hallmark_EMT gene set was strongly inversely correlated with tumor purity. That is, in tumors with higher stromal and/or immune components, there was higher expression of these EMT-related genes.*
- *Expression of the Hallmark_EMT gene set was also highly correlated with expression of a stromal gene set, despite few overlapping genes.*
- *If the EMT-related genes are derived from stroma, this might explain the recurrent observation that expression of EMT-related genes is associated with higher expression of immune genes (i.e., because tumors with lower purity tend to have both higher stromal and immune components).*

Figure 4b indicates a trend towards increased survival. There is multiple testing and no formal statistical plan. Therefore this statement is difficult to justify.

The manuscript has undergone a thorough biostatistical review and revision. Please see the revised statistical methods and analysis and our response to Reviewer 1 comments.

I think the discussion should be shortened. Parts of the discussion should move to the results. The methodology lack some detail. For example the purity experiments.

We have revised the discussion to address the comments raised by the Reviewers and to provide clarification of the key findings of our study. In addition, we have expanded the Methods section to provide additional detail.

REVIEWERS' COMMENTS:

Reviewer #1 (Remarks to the Author):

Overall, the authors have addressed and/or clarified most of my concerns. they provided clarification on many points, discussion has been improved and the paper flows is better. I recommend publication in the current form. A minor point, regarding the font size in the figure (especially in the new ones) please make sure that the font is of reasonably sufficient size in the final document.

Reviewer #2 (Remarks to the Author):

The authors have done an outstanding job addressing the reviewer critiques.

Reviewer #3 (Remarks to the Author):

I think this is much more accurate. I like the work.
Clearly is not possible to tell exactly what is going on due to the differing strands, logistical/tissue challenges and the lack of randomised data. However the translational work had a spectrum of components which suggests stromal EMT does have a role in a resistant phenotype.
The authors should be congratulated. My feeling is that work should be shortened a great deal. The message is straight forward and the written work should reflect this. The figures and stats are much better. Many thanks for changing the work.